# Domain generalization enables general cancer cell annotation in single-cell and spatial transcriptomics

Zhixing Zhong[1,2,8], Junchen Hou[3,8], Zhixian Yao[2,8], Lei Dong [4,8], Feng Liu [5], Junqiu Yue[6], Tiantian Wu [3], Junhua Zheng [2], Gaoliang Ouyang [3], Chaoyong Yang [1,2,7] & Jia Song [2] ✉

Single-cell and spatial transcriptome sequencing, two recently optimized transcriptome sequencing methods, are increasingly used to study cancer and related diseases. Cell annotation, particularly for malignant cell annotation, is essential and crucial for in-depth analyses in these studies. However, current algorithms lack accuracy and generalization, making it difficult to consistently and rapidly infer malignant cells from pan-cancer data. To address this issue, we present Cancer-Finder, a domain generalization-based deep-learning algorithm that can rapidly identify malignant cells in single-cell data with an average accuracy of 95.16%. More importantly, by replacing the single-cell training data with spatial transcriptomic datasets, Cancer-Finder can accurately identify malignant spots on spatial slides. Applying Cancer-Finder to 5 clear cell renal cell carcinoma spatial transcriptomic samples, Cancer-Finder demonstrates a good ability to identify malignant spots and identifies a gene signature consisting of 10 genes that are significantly co-localized and enriched at the tumor-normal interface and have a strong correlation with the prognosis of clear cell renal cell carcinoma patients. In conclusion, Cancer-Finder is an efficient and extensible tool for malignant cell annotation.

It has long been acknowledged that tumor heterogeneity is a significant barrier to the development of effective cancer treatments[1,2]. Single-cell RNA sequencing (scRNA-seq) technology has enabled a comprehensive understanding of intra- and inter-tumor heterogeneity at the level of a single cell, thereby facilitating the development of personalized therapies[3]. Spatial transcriptomics (ST), which follows in the footsteps of scRNA-seq as a promising sequencing technique, captures the spatial context of transcriptional activity within intact tissue[4,5], and is increasingly used in cancer research, resulting in a number of ground-breaking discoveries in the study of cancer heterogeneity[6,7]. In these studies, the precise annotation of malignant state of single cells or spots (measure units in ST) is essential and fundamental.

Currently, malignant cells and spots are identified primarily through marker genes or copy number variation (CNV) events[8]. Clustering and marker gene detection make it possible to identify

[1]Institute of Artificial Intelligence, Department of Chemical Biology, College of Chemistry and Chemical Engineering, Xiamen University, Xiamen 361102, China. [2]Institute of Molecular Medicine, Department of Urology, Renji Hospital, School of Medicine, Shanghai Jiao Tong University, Shanghai 200127, China. [3]School of Pharmaceutical Sciences, State Key Laboratory of Cellular Stress Biology, School of Life Sciences, Xiamen University, Xiamen 361102, China. [4]Department of Pathology, Ruijin Hospital, School of Medicine, Shanghai Jiao Tong University, Shanghai 200025, China. [5]School of Computing and Information Systems, The University of Melbourne, Carlton, Melbourne, VIC 3053, Australia. [6]Department of Pathology, Hubei Cancer Hospital, Tongji Medical College, Huazhong University of Science and Technology, Wuhan 430030, China. [7]Innovation Laboratory for Sciences and Technologies of Energy Materials of Fujian Province (IKKEM), Xiamen 361005, China. [8]These authors contributed equally: Zhixing Zhong, Junchen Hou, Zhixian Yao, Lei Dong. ✉e-mail: songjiajia2010@shsmu.edu.cn

malignant cells/spots, but technical artifacts such as drop-out and high sparsity can lead to false negatives[9]. Furthermore, there is no shared set of cancer-specific marker genes, and the current knowledge of cancer-specific marker genes is insufficient to differentiate malignant cells/spots from normal cells/spots in all tumor microenvironments[10].

As an alternative, copy number variant events can be used to differentiate malignant cells/spots from normal cells/spots in most solid tumors[11]. Representative tools identifying malignant cells/spots in this manner are inferCNV[7,12] and CopyKAT[13]. InferCNV relies on the use of normal cells/spots as a reference and the user's subjective judgment to discriminate malignant cells/spots. On the other hand, CopyKAT must statistically distinguish between intact and aneuploid, making it difficult to obtain high accuracy with high-purity data (a set of nearly exclusively malignant or non-malignant cells)[14]. In addition, current evidence suggests that cell copy number alterations are widespread in normal human tissues[15,16], which can lead to misclassification.

Machine learning, especially neural networks, has introduced additional concepts for automatic cell/spot annotation. In recent years, among the methods for automatic annotation of malignant state based on machine learning, two representative methods for distinguishing the degree of cell malignancy are ikarus[17] and Casee[14]. Ikarus is based on a logistic regression model. Thus, its accuracy varies greatly depending on the selection of the training set, and its generalization performance is poor. Casee, on the other hand, is a transfer-learning-based method for single-cell expression analysis that trains a capsule network classifier using bulk data. This method does not make effective use of the distributional characteristics of single-cell data and its generalization performance may be constrained by the bulk data. Moreover, both of these machine learning methods are currently limited to single-cell data analysis and cannot be applied to ST data. For ST data analysis, deconvolution methods such as cell2location[18] and CARD[19] can determine the cell composition of a spot, but the outcome is highly dependent on reference scRNA-seq data. However, obtaining high-quality reference datasets can be difficult, posing a substantial obstacle to relevant research[20,21]. To accomplish this task, a reference-free malignant cell annotation algorithm with high accuracy, good generalization performance, and easy scalability to handle multiple data types is urgently required.

In this work, to address the issues described above, we present Cancer-Finder, a domain generalization-based malignant cell annotation strategy that can learn a generalization model from multiple datasets with varying distributions. This allows direct distinction of malignant and normal cells in the pan-cancer tumor microenvironment within single-cell data with undefined distribution (unknown domain). In addition, by substituting the training set, we rapidly extend Cancer-Finder to annotate malignant spots in ST data and demonstrate its high prediction accuracy after training with a small training set. By precisely identifying malignant spots on 5 ccRCC ST slides, we successfully identify a gene signature consisting of ten genes that tends to be enriched at the interface between tumor and normal tissue, may be associated with the formation of an invasive tumor microenvironment, and serves as a desirable prognostic indicator. Our data suggest that Cancer-Finder is an efficient and extensible tool for annotating cellular or spatial-spot states and will facilitate the discovery of biological mechanisms using single-cell and ST data.

## Results

### Domain generalization enables general malignant cell identification in the tumor microenvironment

For tumor microenvironment research using scRNA-seq or ST data, accurate cell annotation is required, specifically, accurate annotation of malignant status. Unfortunately, cell expression profiles from different tissues have varying distributions, complicating generalization of classification models trained on one dataset to other datasets[22]. The

field of machine learning has recently made progress towards the goal of domain generalization, which enables learning knowledge and training models from a variety of known domains (training domains) and generalizing to unknown but similar domains (testing domains)[23], where data from different domains are considered to follow different distributions. Typically, a "domain" refers to the specific data type or category on which a model is trained. Here, we apply this concept to the annotation of cellular malignant states in single-cell or spatial data by assuming that data from different tissues arise from different domains. The generalized model based on multiple domains is trained to predict labels for test data from different domains (including unknown domains). This led to the development of Cancer-Finder.

A deep neural network consisting of a feature extraction module and a classification module is proposed for this purpose (detailed in Methods). As shown in Fig. 1, the feature extraction module comprises two fully connected layers, separated by a random dropout layer to prevent overfitting. The feature extraction module is connected to a classification module which consists of a classification layer to complete the classification task. The number of neurons in the classification layer corresponds to the number of classes required for the classification task, which in this case, is two. The output scores of the classification layer, namely $g^{malignant}$ and $g^{non-malignant}$, are employed for the discriminative inference of malignant cells. A softmax value, ranging from 0 to 1, is calculated based on these two scores to differentiate between malignant and non-malignant cells, with a default threshold of 0.5. The network is trained with single-cell or spatial expression profiles from multiple tissues, and it can then predict the state of new data.

Empirical risk is defined as the value of the loss or discrepancy between the predicted and real labels across the training dataset. Empirical Risk Minimization (ERM) is an important principle in machine learning[24]. Proposed on the basis of empirical risk, Risk extrapolation, a type of robust optimization across a perturbation set of extrapolated domains, can reduce a model's sensitivity to a broad range of distribution shifts[25]. Therefore, a loss function optimization method based on risk extrapolation is employed for domain generalization in this study. Detailly, we define the empirical risk of a domain (i.e., a tissue) as the cross-entropy of the predicted and real labels of objects from the domain. Risk extrapolation then utilizes two types of global risks, variance risk and average risk, to evaluate the performance of the model across multiple domains. The variance risk is calculated as the variance of risks across different training domains to reflect the disparity of training risk across domains. Alternatively, the average risk is calculated for data across all training domains to reflect the total training risk across domains (details are provided in Methods). To achieve a model with good performances in all domains, both variance risk and average risk must be minimized. Additionally, during training, nodes in the first hidden layer are dropped at random to prevent overfitting.

In this way, the resulting pre-trained model in Cancer-Finder can be used to successfully extract and transform features from scRNA-seq and ST data of new samples, infer the malignant states of cells, and visualize the cells directly according to their malignant state for cancer data (Fig. 1). Furthermore, to enhance the model's interpretability, Cancer-Finder integrated an interpretability module that utilizes a modified saliency map[26] (details are provided in Supplementary Note 1). Thus, Cancer-Finder can also be utilized to investigate key features that differentiate malignant cells from non-malignant cells, serving as a valuable tool for exploring cancer-related mechanisms.

### Cancer-Finder enables efficient malignancy annotation for scRNA-seq from multiple tissues

We collected 74 human tumor microenvironment datasets from Tumor Immune Single Cell Hub (TISCH)[27] as training sets and then

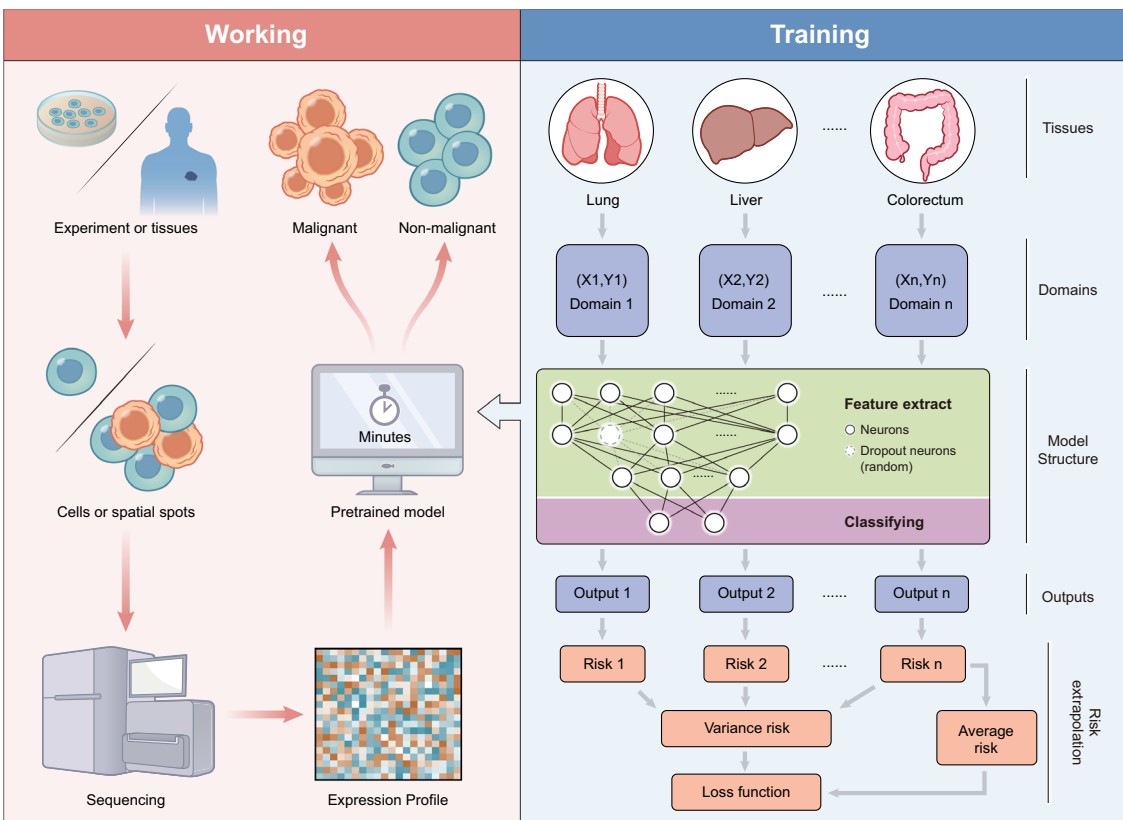

**Fig. 1 | Overview of Cancer-Finder and its application.** Cancer-Finder is a scalable framework that uses single-cell sequencing data to accurately annotate the malignant status of cells and is easily extensible to annotate other data (e.g. ST). The pretrained model accurately and quickly identifies malignant cells derived from cancerous tissues. To counteract differences among different tissues, Cancer-Finder employs a domain generalization training strategy to improve general discriminatory performance and accurately identify malignant cells in unexplored domains. Typically, a "domain" refers to the specific data type or category on which a model is trained. Here, we apply this concept to the annotation of cellular malignant states in single-cell or spatial data by assuming that data from different tissues arise from different domains. The model is a neural network composed of an input layer and two hidden layers for feature extraction and a layer for classification. Cancer-Finder uses risk extrapolation for domain generalization. This optimizes the model for high accuracy in all tissues because reducing risk differences across training domains can decrease a model's sensitivity to a broad range of distribution shifts. In order to evaluate the performance of the model across multiple domains, this method minimizes two types of global risks: variance risk and average risk.

organized them into 14 distinct categories according to their original tissues. The cell malignancy annotations from the database served as training labels for the collated cells, with down-sampling employed to ensure balanced categories. In total, 328,230 single-cell expression profiles from 13 distinct tissues were utilized after data preprocessing (Fig. 2a, Supplementary Note 2). The data were separated into a training set of 80% and an internal validation set of 20%. Then, to prevent over-fitting of noise within the TISCH database, we used normalized breast cancer data from the original CopyKAT[13] study as extra-domain validation data. Only the model whose accuracy of extra-domain validation reaches its maximum was kept for use (Supplementary Fig. 1).

Notably, it is more difficult to distinguish malignant cells from healthy cells than it is to identify other cell characteristics, such as cell types. t-SNE plots, based on highly variable genes, indicated that cells from different tissues are more easily distinguished than cells from different malignant states (Fig. 2b, c). However, the distribution of cells on the newly generated t-SNE plot appears consistent with the malignant nature of the cells after feature extraction and transformation using Cancer-Finder (Fig. 2d), demonstrating Cancer-Finder's powerful feature selection ability.

To evaluate the performance of Cancer-Finder for annotating malignant states, we monitored the changes in risks (variance risk and average risk, the two main components of the loss function) and accuracy throughout the training process. As shown in Fig. 2e, the variance risk increased rapidly and then gradually decreased to a steady state. This indicates that the model transitions from a random initialization state to a tissue-specific adaptation state, and then gradually evolves into a state of malignant commonality across tissues. Simultaneously, the average risk as a regular term decreases steadily, leading to an improvement in the overall accuracy of malignancy classification (Fig. 2f). After 50-70 training epochs in 5 repetitions, the accuracy of the internal validation dataset reached a steady state (Fig. 2g), as did the accuracy of the extra-domain validation dataset (breast cancer data). On all 13 tissues in the internal validation dataset, the average accuracy of these five pre-trained Cancer-Finder was 95.16%, with the average accuracy of malignant status prediction in 11 tissues exceeding 90% (Fig. 2h).

To externally test the accuracy of this pre-trained model, we utilized a scRNA-seq dataset of peripheral blood mononuclear cells (PBMC) from the official website of 10x genomics (defined as dataset 1, detailed in Supplementary Tables 1, 2) and cancer cell lines from a previous study[28] (defined as dataset 2, detailed in Supplementary Tables 1, 2), both of which had known malignant status labels. Five repetitions yielded an average accuracy of 98.30% for the pre-trained model (Fig. 2h, The methodology of the repetition is detailed in Supplementary Note 2). Above all, using our training dataset, internal validation set, and external validation set, we validated the robustness of Cancer-Finder in identifying malignant cells.

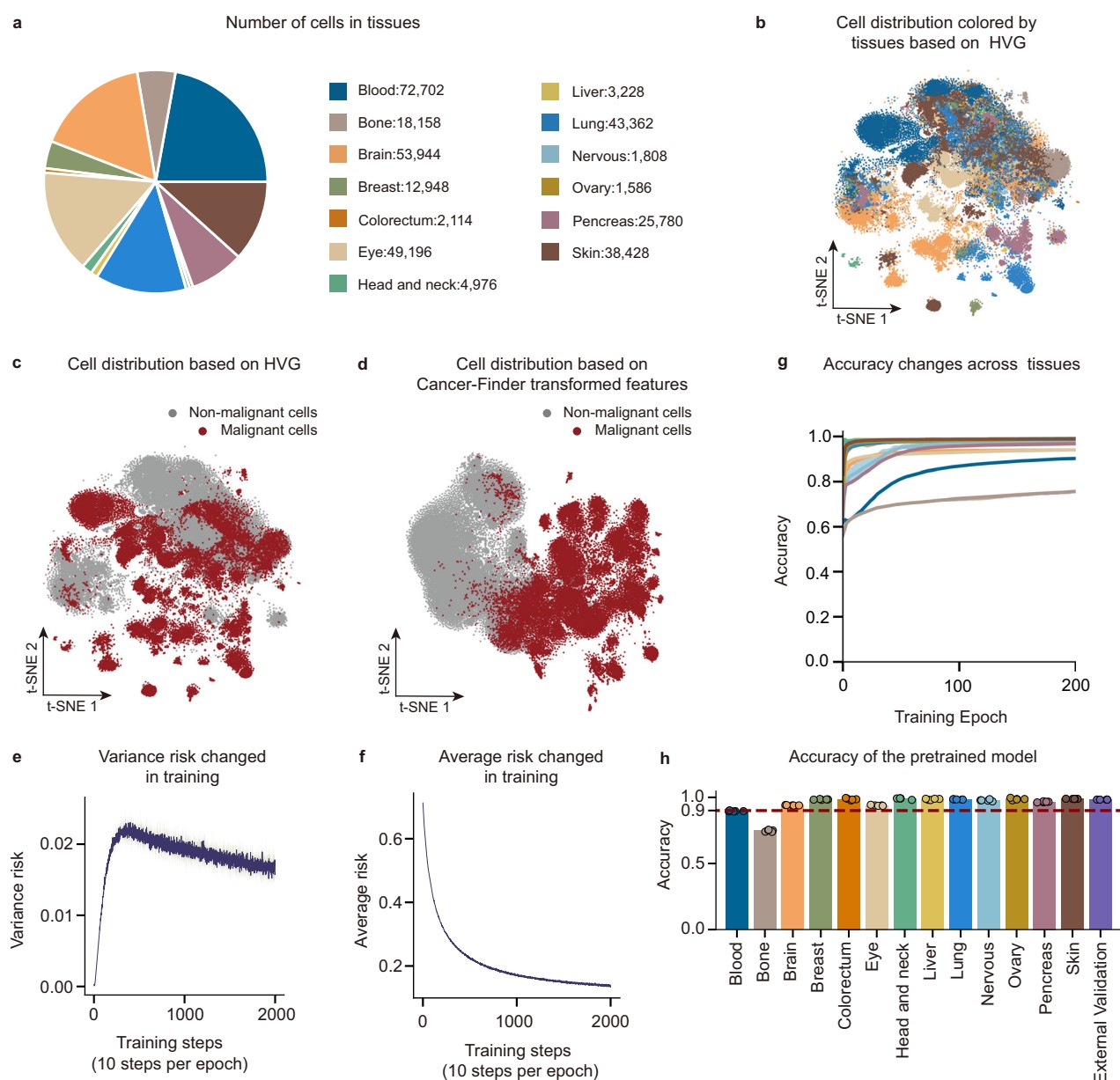

**Fig. 2 | Performance evaluation of Cancer-Finder. a** Training data structure. In this study, 328,230 cells from 13 distinct tissues were finally utilized. **b,** Visualization of internal validation data from 13 tissues utilizing t-SNE based on highly variable genes, wherein each point on the graph represents a single cell, each color represents a tissue, and each tissue has the same color as in **a**. **c** Similar to **b**, the t-SNE visualization of data from 13 tissues by highly variable genes, colored according to malignant status. Red represents malignant cells and gray represents normal cells. **d** t-SNE visualization utilizing highly-variable, neural network-transformed features and colored according to malignant status. Red represents malignant cells and gray represents normal cells. **e** Changes in average risk throughout the training process. The training was carried out five times. **f** Changes in variance risk throughout the

training process. The training was carried out five times. **g** Changes in the accuracy of prediction for different datasets from different tissues (13 tissues, using the same colors as in **a** throughout the training process. **h** Accuracy of the pre-trained models. Accuracy of each tissue, including internal validation data for 13 tissues and external validation data (mixed cell lines and PBMC, colored in purple). In the data presented in **e**–**h**, we conducted five training sessions for Cancer-Finder, completing 5 independent and repeated experiments. In detail, 5-fold leave-cells-out cross-validation was applied. The total sample size used for internal validation is 328,230 cells ($n = 328,230$), and for external validation, the sample size is 15,986 cells ($n = 15,986$). In each independent cross-validation experiment, 65,646 cells ($n = 65,646$) were examined as the internal validation experiments.

## Performance comparison with existing methods on scRNA-seq datasets

In addition to the PBMC dataset and malignant cell line dataset (defined as dataset 1 and dataset 2, respectively, and used as 2 gold standard datasets), we also collected eight additional scRNA-seq datasets from a variety of tissues to serve as silver standard external test data (defined as dataset 3-10, detailed in Supplementary Tables 1, 2). These eight silver standard datasets include medulloblastoma cells[29], circulating tumor cells (CTCs)[30,31], hepatoblastoma cells[32], head

and neck cancer cells[33], ovarian, colorectal, lung and breast cancer cells[34], and were used to evaluate Cancer-Finder's performance in annotating pan-cancer cells in comparison to other available tools (Fig. 3a). The annotation from the original study is used as true labels (detailed in Supplementary Table 2). These 10 datasets (2 gold standard datasets and 8 silver standard datasets) had varying malignant cell percentages and ranged from a minimum cell count of 357 to a maximum cell count of 93,575 to simulate a variety of clinical scenarios (Fig. 3a).

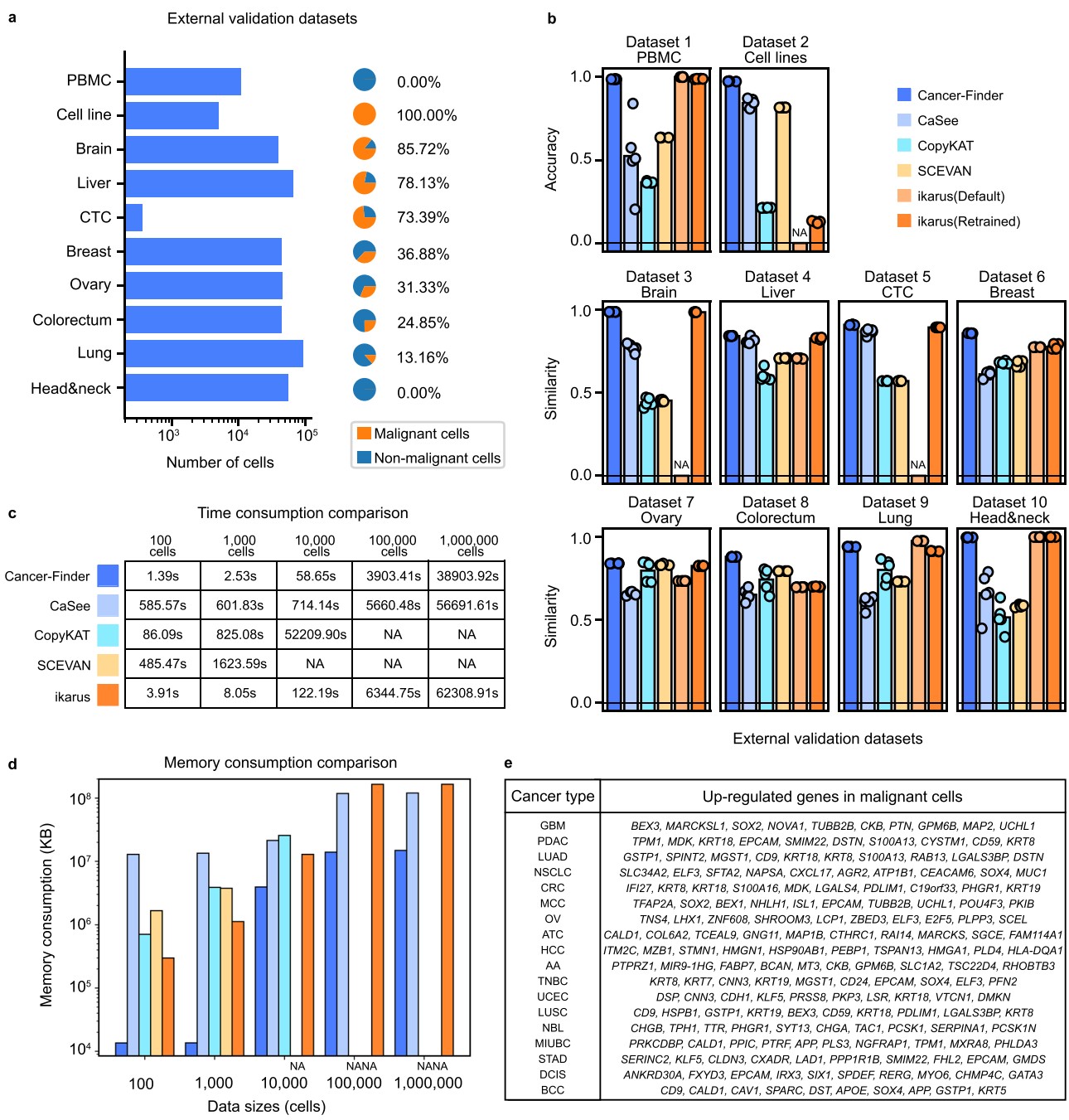

**Fig. 3 | Performance comparison with existing methods based on external validation datasets and application to a large database. a** Data structure of ten external validation datasets with varying cell counts and proportions of malignant cells. Among them, the head and neck dataset exclusively concentrated on immune cells, with malignant cells being experimentally excluded. **b** Comparison of Cancer-Finder's prediction accuracies to four other cell annotation algorithms on 10 external validation datasets. Cancer-Finder demonstrated significantly greater accuracies and similarities (to the labels from the original studies) than other tools across various cancers. Since most of these available algorithms exhibit some level of randomness in their results across runs, all tests were conducted in parallel five times. It is noteworthy that the pre-trained Cancer-Finder consistently yields uniform predictions on the external datasets. Recognizing that variations in the training process and data may introduce a degree of randomness, we conducted five training sessions for Cancer-Finder here, completing the specified five independent and repeated experiments (detailed in Supplementary Note 2). The detailed cell numbers (*n* numbers) and malignancy percentages for each dataset are shown in **a**. For ikarus (retrained), we employed the same strategy. The presence of an 'NA' denotes that the method returns an error and cannot be executed with these data. **c** Comparison of Cancer-Finder's inference speed to the other four methods. 'NA' indicates that the method could not run correctly on the data. **d** Comparison of Cancer-Finder's memory consumption to the other four methods. 'NA' indicates that the method could not run correctly on the data. This study utilized a maximum memory size of 512 G Bytes. **e** Up-regulated genes identified from predicted malignant cells. Full names of the cancers are detailed in Supplementary Table 15. Gene names are formatted in italics. Source data are provided as a Source Data file.

We compared Cancer-Finder to four other tools, namely SCEVAN[35], CopyKAT[13], CaSee[14], and ikarus[17] (see Supplementary Table 3 for detailed information). For ikarus, we evaluated both the model provided in its original study and a model that was retrained using our training set. As a measure of accuracy, the accuracy rates were calculated for the gold standard data, whereas for silver standard data, the similarity rates to the true labels were used (the difference between accuracy and similarity is detailed in Supplementary Note 3). As shown in Fig. 3b, Cancer-Finder achieved an overall accuracy of 98.30% for the cells from the gold standard datasets, outperforming all other methods (SCEVAN: 68.71%, CaSee: 62.39%, CopyKAT: 32.21%, ikarus: 68.72%, the retrained ikarus: 71.92%). Each tool was evaluated five times with the same parameters on each dataset and detailed accuracies of each dataset in each run are shown in Supplementary Tables 4, 5. Cancer-Finder also demonstrated greater similarities with the original annotated labels for the cells in the silver standard datasets, achieving an overall average similarity of 90.89% (Fig. 3b, SCEVAN: 67.98%, CaSee:67.89%, CopyKAT:64.68%, ikarus:75.04%, the retrained ikarus: 86.88%). Detailed similarities of each silver standard dataset in each run are shown in Supplementary Tables 6–13. Additional algorithmic performance evaluations, including precision, recall, F1 scores, the area under the ROC curve (AUROC) and the area under Precision-Recall (PR) curve (AUPRC) values, are detailed in Supplementary Figs. 2–5 and Supplementary Data 1. These results further demonstrate Cancer-Finder's superior performance.

More specifically, CopyKAT, a representative method based on CNV inference, showed significantly decreased accuracy in datasets with a high percentage of malignant cells. CaSee, a method of transductive learning, requires training on bulk data, limiting its accuracy on single-cell datasets and exhibiting instability in data with a low proportion of malignant cells. The pre-trained model and selected feature set provided by ikarus are constrained by the training data and difficult to generalize to other datasets. Furthermore, it was impossible to provide results on three datasets (datasets 2,3 and 5). The performance of ikarus was improved after being retrained using the same large-batch pan-cancer data as Cancer-Finder, but its performance on the cell line dataset remained inferior. This may be due to the use of the logistic regression model, which lacks predictive power for unknown domains, such as the cell line dataset, whose similar data were absent from the training set. In comparison to these four currently available algorithms, Cancer-Finder exhibits stable and superior performance on datasets having varying sizes, malignant cell percentages, and domains (whether or not they are included in the training set), indicating its promising application.

### Application to a large database
Additionally, Cancer-Finder offers significant inference speed advantages via the use of direct inference with pre-trained models. We tested the inference speed of Cancer-Finder on datasets of varying sizes, including 100, 1000, 10,000, 100,000 and 1,000,000 cells (detailed in Supplementary Note 4), and found inference times of 1.39 s, 2.53 s, 58.65 s, 3,903.41 s and 38,903.92 s, respectively, on a computer with 16 cores, 2.1Ghz CPU and NVIDIA 3090 GPU. These times were superior to those of other methods (Fig. 3c, detailed in Methods). Furthermore, we discovered that the majority of the time required for large datasets was spent on data reading due to the 'csv' text format for storing the expression matrices to allow for fair comparisons to other methods. In additional tests, use of binary-stored matrices allowed Cancer-Finder to infer 10,000 cells in 4.15 s and 100,000 cells in 39.46 s (detailed in Supplementary Note 4). Simultaneously, it is noteworthy that Cancer-Finder exhibits a significantly reduced memory consumption compared to other algorithms (detailed in Fig. 3d and Supplementary Table 14), indicating its potential for future analysis of large-scale datasets.

Thererfore, Cancer-Finder was applied to annotate scRNA-seq data from Cancer Single-cell Expression Map (CancerSCEM)[36] database. Over 500,000 cells can be quickly predicted by Cancer-Finder in one hour. Since CancerSCEM only provides the percentage of malignant cells on each dataset and does not provide annotation information for each individual cell, we compared the percentage of malignant cells predicted by Cancer-Finder with the percentage provided by the database. The two are highly correlated (Pearson's correlation coefficient > 0.85, $p < 0.0001$, $n = 152$, two-tailed test). In conjunction with Cancer-Finder's cross-validation results on data of TISCH (Supplementary Fig. 6), these results indicate that Cancer-Finder can obtain highly consistent results with complex annotation strategies (combining both manual and algorithmic annotations), demonstrating its accuracy.

On the basis of the annotation results, we also performed differential gene expression analyses between malignant and non-malignant cells of each cancer to identify candidate marker genes. The results are presented in Fig. 3e (full names of the cancers are detailed in Supplementary Table 15), where it can be seen that the marker candidates identified using this strategy are consistent with available knowledge. For instance, *EPCAM, KRT8, KRT18* are widely employed markers for malignant cell or cancer stem cell across various malignancies such as cervical cancer[37], breast cancer[38] and head & neck cancer[39], and *SOX2* was considered as a cancer markers in glioblastomas[40] and esophageal squamous cell carcinoma[41].

These results indicate that Cancer-Finder may be a useful tool for cellular annotation during database development, thereby reducing the time-consuming manual annotation process.

### Expanding Cancer-Finder to spatial transcriptomics annotation
ST sequencing technology, such as 10x Visium spatial transcriptomics, uses barcoded spots with a diameter of 55-100 um to capture cells in situ for sequencing, which may include multiple cells per spot. The identification of spots containing malignant cells is crucial for cancer research and can assist cancer researchers in locating cancerous regions for the study of the tumor microenvironment.

Cancer-Finder uses models that have been pre-trained using training data for inference and can perform a variety of inference tasks by substituting training data. By replacing scRNA-seq data in the training set with spssatial transcriptomics, Cancer-Finder offers an approach to directly and rapidly infer the location of malignant cells on the spatial slides without reference, thereby demonstrating a good scalability (Fig. 4a).

A total of 14 Visium spatial slides were collected, including breast cancer (BRCA), hepatocellular carcinoma (HCC)[42], intrahepatic cholangiocarcinoma (ICC)[42], colorectal cancer (CRC)[43], ovarian cancer (OV) and renal call carcinoma (RCC)[44] (detailed in Supplementary Table 16). The classification of non-malignant and malignant spots was referenced from a previous study[45] and revised by pathologists, and some slides without available annotation were manually annotated by pathologists directly. As shown in Fig. 4b, c, when the model was trained with a small amount of spatial transcriptomics data, its predictions for the spatial data of the cancers it was trained on were highly consistent with the pathologist's annotations (accuracy: 82.00-97.37%). The overall performance of Cancer-Finder is relatively good, but the accuracy of some untrained cancers is less desirable. This may be due to the fact that the overall training dataset is still limited. As data accumulates, we will collect additional data for future training and updates.

Moreover, the evaluation of Cancer-Finder on spatial transcriptomics (ST) data generated by other ST techniques (MERFISH[46], Slide-seq[47], legacy ST[48]) was carried out. Notably, other platforms have fewer use cases than the widely used commercial platform 10x Visium, and even fewer instances of cancer tissue data, making it difficult to collect enough data for model training. Thus, models trained on single-

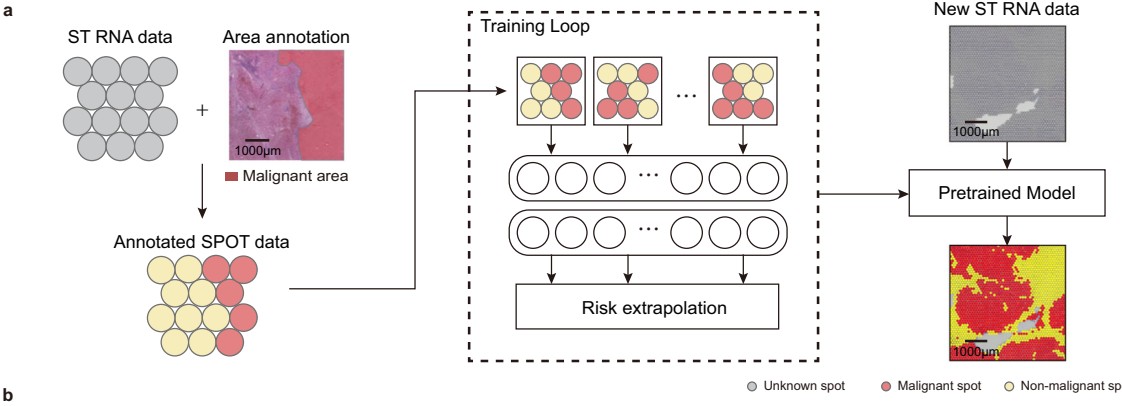

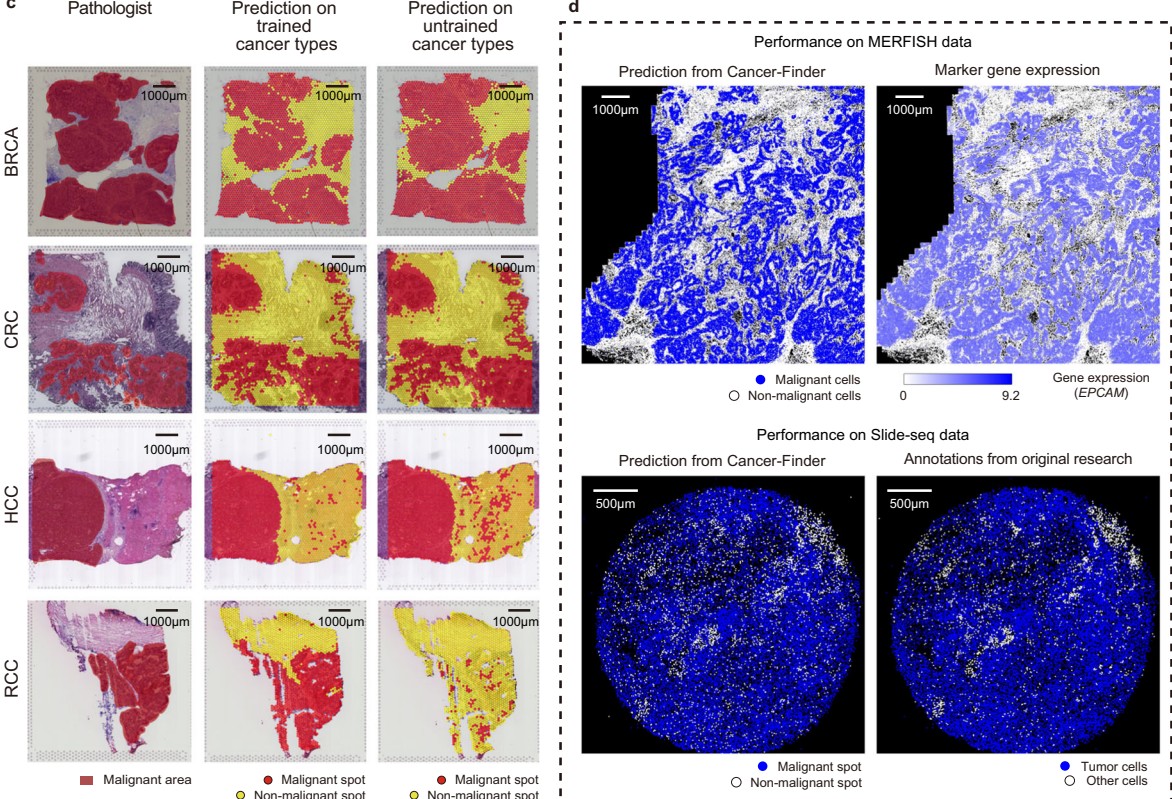

**Fig. 4 | Expansion of Cancer-Finder to spatial transcriptomic annotation.**
**a** Expansion of the training process to include ST data. Annotations from pathologists were used to determine which spots were malignant. Training is completed by replacing the single-cell matrix with the spot expression matrix directly.
**b** Specifics regarding the datasets used during spatial transcriptome training and testing, as well as the accuracy rates obtained. **c** Comparison of model predictions

and annotation labels (left) from pathologists for cases where the tested cancer type was included (middle) or excluded (right) from the training data.
**d** Comparison between model predictions and expression of malignant marker genes on MERFISH data (up), and comparison between model predictions and annotations from original research on Slide-seq data (down). Gene names are formatted in italics. Source data are provided as a Source Data file.

cell or 10x Visium datasets are applied to these data[49,50]. The results suggest that existing models are easily adaptable to datasets with comparable sequencing resolution (Fig. 4d). Additional details are available in Supplementary Fig. 7 and Supplementary Note 5.

These results indicate that Cancer-Finder can be easily expanded to include the annotation of ST data. Moreover, after training on a small collection of training data, Cancer-Finder is able to make predictions with high accuracy. Thus, as the quantity of ST grows and the demand for related research increases, there is great potential for future applications of Cancer-Finder in this area.

### Application of Cancer-Finder on intertumor heterogeneity analysis in ccRCC ST dataset

The tumor immune microenvironment (TME) is an intricate and complex system that plays a crucial role in the development of tumors. It consists of numerous cell types, such as immune cells, stromal cells, and extracellular matrix components[51]. The composition of cells within the ccRCC TME and their interactions are believed to be significant factors influencing patient outcomes[52,53]. In this study, we expect to begin with Cancer-Finder's spot status prediction, determine tumor and histologically normal tissues on ST sections based on the prediction results, investigate significant features that differentiate malignant and non-malignant spots, and investigate the relationship between these features and prognosis to provide insight into the TME of ccRCC.

Five ccRCC 10x Visium slides containing tumor and histologically normal tissues adjacent to the tumors from a previous study were analyzed with Cancer-Finder (detailed selection criteria is in Methods)[54]. As demonstrated in Fig. 5a, classical ccRCC markers like *CA9*[55] and *ANGPTL4*[56] failed to show consistency when determining malignant spots. This may cause confusion in downstream analysis. In contrast, Cancer-Finder presented a robust prediction of malignant spots, which was highly in line with the pathologist's manual annotation and is mostly consistent with the results of the single-cell reference-based deconvolution[19] (Fig. 5a). Meanwhile, the interpretability module of Cancer-Finder ranked the importance of features and identified the top ten genes (*NDRG1, TAGLN, MALAT1, IGKC, IGHA1, IGHG4, IGLC2, IGHG3, SOD2* and *KRT19*, detailed in Supplementary Note 1 and Supplementary Fig. 8). These genes are believed to play a significant role in differentiating malignant and non-malignant spots.

To investigate the spatial expression patterns and functions of these genes further, single sample Gene Set Enrichment Analysis (ssGSEA)[57] was used to calculate an expression signature score for these genes on the ST slides. Interestingly, this signature tends to be enriched in the tumor-normal interface (Supplementary Fig. 9). By analyzing the expression of these ten genes on each spot, it was determined that spots in which nine of these genes were expressed (defined as co-localization state) were also located at the tumor-normal interface (Fig. 5b). On the basis of cell deconvolution, spots with co-localization states were analyzed to determine the major cell types contributing to this signature. The proportions of the top ten representative cell types in these 5 slides are shown in Fig. 5c. Cell types that might be attributed to these genes are displayed in Fig. 5d. Immunoglobulin-related genes (*IGKC, IGHA1, IGHG4, IGLC2, IGHG3*) were mainly the byproducts of plasma. *TAGLN* was mainly expressed in fibroblast and has been proven to play a significant role in EMT process[58] and RCC invasion[59]. *MALAT1* and *SOD2* were universally expressed across all these cell types while *KRT19* demonstrated higher expression in cancer and Ascending Thin Limb of Loop of Henle cells (LoH ATL cells). These genes were also active participants in the epithelial-mesenchymal transition (EMT) process in various cancers[60–62]. Given the hints above, we further investigated the potential prognostic value of this signature. 530 bulk RNA-seq samples from The Cancer Genome Atlas (TCGA)[63] were analyzed and ssGSEA signature scores for patients were inferred and compared with the

Hallmark EMT pathway and the ccRCC EMT program[54]. As shown in Fig. 5e–g, the signature revealed by Cancer-Finder can serve as a better prognostic indicator than these available EMT programs. These findings deserve further mechanistic studies, which could lead to a better understanding of the renal cancer microenvironment.

In conclusion, Cancer-Finder has superior or comparable capabilities in identifying malignant spots and does not require a priori information and data compared to conventional markers or deconvolution techniques. Simultaneously, the most significant features from the model interpretability module can aid in the study of important gene expression patterns and their related cells in TME, and can be combined with prognostic assessment to generate prognostic indicators. The aforementioned characteristics demonstrate the considerable potential of Cancer-Finder for application in studies associated with TME.

## Discussion

On the assumption that, deep learning has the potential to learn a generally accurate rule from a mostly accurate training set, we have developed a domain generalization (DG) approach (Cancer-Finder) to the task of malignant state annotation that can effectively annotate pan-cancer scRNA-seq and ST data. Due to its high performance and computational simplicity, risk extrapolation is employed here (a detailed discussion is in Supplementary Note 6 and Supplementary Table 17). Combining average risk and variance risk in risk extrapolation enables Cancer-Finder to achieve a good generalization performance across datasets (Supplementary Fig. 6c, d), cancer types (Supplementary Fig. 6a, b), and technology platforms (Supplementary Fig. 10). Additionally, it is robust to a certain percentage of mislabeling in training set (Supplementary Fig. 11).

Compared to existing techniques, Cancer-Finder established greater precision and stability in malignant annotation on scRNA-seq datasets derived from a variety of cancers, achieving an accuracy of 98.30% in golden standard datasets and a similarity of 90.89% in silver standard datasets (The majority of the prediction errors may be due to low sequencing quality in some cells, a problem that can be resolved by increasing the sequencing depth, Supplementary Fig. 12). Cancer-Finder is more accurate because deep learning models are more adaptable and have a greater capacity for fitting than traditional models such as logistic regression[64,65]. In addition, Cancer-Finder makes effective use of accumulated cancer tissue data and annotation information (primarily through algorithmic calculations and manual annotations), thereby increasing the chance of accurately distinguishing between malignant and non-malignant cells. While the majority of existing algorithms are based on simple models or single-dataset analyses, the former is susceptible to model limitations, whereas the latter is susceptible to the quality of the focused dataset and the cell type it contains. Unlike other methods (CopyKAT must infer CNV and classify based on CNV profiles, SCEVAN needs to characterize the clonal structure and CaSee must find a reference to train), the inference process of Cancer-Finder requires only a simple forward-propagating linear computation. As the amount of single-cell data increases, we believe that retraining with larger amounts of data will afford Cancer-Finder considerable potential in cancer research[66].

By replacing the training dataset, we quickly extended Cancer-Finder to annotate malignant spots in the ST data and demonstrated its ultra-high prediction accuracy after training with a small training set. Despite having only a small number of relevant ST data in the training set, Cancer-Finder displays a high level of accuracy (82.00-97.37%) on ST data of trained tissues. In addition, the pre-trained Cancer-Finder can be easily extended to ST datasets generated by other techniques with comparable sequence resolution, validating Cancer-Finder's great generalization capability (Supplementary Fig. 7).

In addition to the expansion of training data types, Cancer-Finder can also be expanded to annotate other cell states (or cell

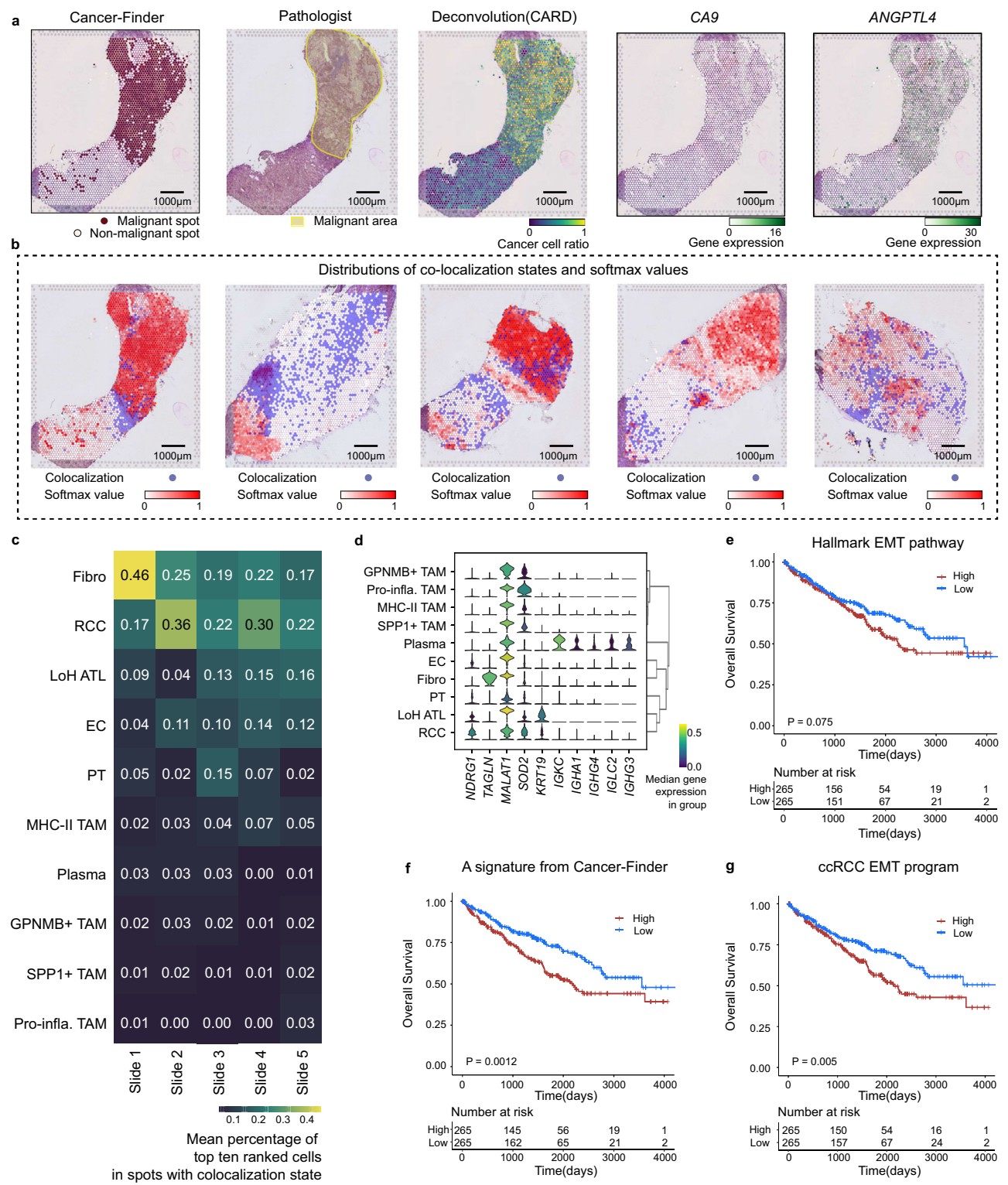

**Fig. 5 | Application of Cancer-Finder on intertumor heterogeneity analysis in ccRCC ST dataset. a** Annotations of malignant regions obtained by different methods, including prediction by Cancer-Finder (first left), annotation by pathologists (second left), deconvolution by CARD (middle), gene expression of *CA9* (a marker of ccRCC, second right) and gene expression of *ANGPTL4* (a marker of ccRCC, first right). **b** Co-localization of genes in the gene signature (blue) and prediction of malignant area (red). The co-localization state was assigned to the spot when nine of the ten genes in the signature have expression values (UMI count > 0). The result of malignancy prediction is represented by the softmax value, with the likelihood of being malignant increasing as the value rises. Where 0.50 is the default softmax cutoff value. **c** The proportions of the top ten representative cell types in spots with a co-localization state across 5 slides. **d** Gene expression of genes in the detected gene signature across cells. **e** Survival analyses using the ssGSEA score of the genes of Hallmark EMT pathway, a two-sided log-rank test was applied ($n = 530$, $p = 0.075$). **f** Survival analyses using the ssGSEA score of the gene signature from Cancer-Finder, a two-sided log-rank test was applied ($n = 530$, **$p = 0.0012 < 0.01$). **g** Survival analyses using the ssGSEA score of the genes of ccRCC EMT program, a two-sided log-rank test was applied ($n = 530$, **$p = 0.005 < 0.01$). Gene names are formatted in italics. Source data are provided as a Source Data file.

types) by substituting training labels. Changing the training label to immune cells, for example, allows Cancer-Finder to accurately identify immune cells from single cells. In our external tests, Cancer-Finder's accuracy in identifying immune cells in the lung, breast, ovary, and liver ranged from 85.21% to 95.76% (detailed in Supplementary Fig. 13). As single-cell data accumulates, we will be able to use Cancer-Finder for annotation of a variety of cell states (or types), such as rare cells (which are too rare to be detected in a single sample or are difficult to cluster into groups, making annotation difficult).

Despite these advantages, the current model still has room for development. Cancer-Finder performs well (accuracy > 0.8) on most cancers, but its performance is limited on hematologic tumors (Supplementary Fig. 14), possibly due to the significant difference between hematologic and solid tumors. We do not therefore recommend Cancer-Finder for hematologic tumor data. Moreover, Cancer-Finder overlooks the spatial relationships among spots in ST data, a factor with the potential to enhance its overall efficacy. All of these merit further exploration.

## Methods

### Design and implementation of Cancer-Finder
Cancer-Finder is built on Python and the Pytorch deep-learning framework, providing users with the ability to quickly identify malignant cells on scRNA-seq, ST slides or extended data types. The model accepts the normalized expression data. For raw count data, log-transformed global-scaling normalization is required, which is accomplished by log-transforming the normalized result from the 'NormalizeData' function of Seurat[67] or 'pp.normalize_total' function of SCANPY[68]. The pre-trained models mentioned in this article are available for download, allowing for convenient and rapid inference on scRNA-seq and ST data. In addition, scripts are provided to facilitate retraining for new discrimination tasks. Both CPU-only and GPU platforms are supported for execution.

### Structure of the model
**Feature extraction and classification.** Cancer-Finder's model consists of a feature extraction module and a classification module. Feature extraction module comprises two fully connected layers, separated by a random dropout layer to prevent overfitting[69]. The default dropout probability for the intermediate dropout layer was set to 0.5, as this value was recommended[70], and commonly employed in similar studies[71,72]. Without the use of activation functions, the layers are connected directly. The first of fully connected layers employs the same number of neurons as the selected features after data processing (4572 for scRNA-seq data, 5000 for ST data), while the second fully connected layer reduces the number of neurons to 512, creating a bottleneck layer. The feature extraction module is connected to a classification module which consists of a classification layer to complete the classification task. The number of neurons in the classification layer corresponds to the number of classes required for the classification task, which in this case, is two. The output scores of the classification layer, $\mathbf{g} = (g_1, \ldots, g_c)$, is used for the discriminative inference of malignant cells. On the basis of $\mathbf{g}$, a softmax value ranging from 0 to 1 is calculated to distinguish between malignant and non-malignant cells.

The formula for calculating the softmax value is:

$$\text{Softmax} = \frac{\exp\left(g^{\text{malignant}}\right)}{\sum_{i=1}^{c} \exp\left(g^i\right)} \tag{1}$$

where $c$ is the number of classes, $g^{\text{malignant}}$ is the output score of the malignant class. In this study, malignant cells or spots are predicted when the softmax value is greater than a default threshold of 0.5 when $c = 2$.

**Training approaches for domain generalization tasks.** The risk functions used in the training process of domain generalization were proposed by David Krueger et al.[25] and named risk extrapolation. This method serves two purposes: 1) to reduce training risk, and 2) to increase the similarity of training risk across domains to complete domain generalization. In this study, data from m different tissues in the training set are considered as different source domains and denoted as $x_1, x_2, \ldots x_m$. The data labels are denoted as $y_1, y_2 \ldots y_m$, and the feature extraction and classification network is denoted as $f$. Risks of $m$ training source domains are represented as $\mathcal{R}_1, \ldots, \mathcal{R}_m$. The cross-entropy loss function is used to represent the training risk of each domain $x_e$, for any $e \epsilon \{1, 2, \ldots, m\}$:

$$\mathcal{R}_e(\boldsymbol{\theta}) = \text{CrossEntropy}\left(f_{\boldsymbol{\theta}}(x_e), y_e\right) \tag{2}$$

To prevent excessively high values for the total training risk from multiple source domains, we use the mean of the domain risks $\mathcal{R}_{\text{avg}}$ to characterize the training risk:

$$\mathcal{R}_{\text{avg}}(\boldsymbol{\theta}) = \frac{1}{m} \sum_{e=1}^{m} \mathcal{R}_e(\boldsymbol{\theta}) \tag{3}$$

To describe the discrepancy of training risk across domains, we use the variance of training risks across training domains as follows:

$$\mathcal{R}_{\text{var}}(\boldsymbol{\theta}) = \frac{\sum_{e=1}^{m} \left(\mathcal{R}_e(\boldsymbol{\theta}) - \mathcal{R}_{\text{avg}}(\boldsymbol{\theta})\right)^2}{m} \tag{4}$$

The loss function, which is proposed to reduce training risk while increasing the similarity (reducing the discrepancy) of training risk across domains, is described as follows:

$$L(\boldsymbol{\theta}) = \beta^* \mathcal{R}_{\text{var}}(\boldsymbol{\theta}) + \mathcal{R}_{\text{avg}}(\boldsymbol{\theta}) \tag{5}$$

The first term of the equation controls the domain-to-domain risk variance, whereas the second term controls the total training risk. $\boldsymbol{\theta}$ represents the model parameter and $\beta$ is a hyperparameter to adjust the weights of the two risk terms. Detailly, $\beta$ serves as an important hyperparameter in the risk extrapolation method, controlling the balance between reducing the average risk and enforcing equality of risks, with $\beta \to 0$ recovering ERM, and $\beta \to \infty$ leading to a complete concentration on making the risk equal, thereby completing domain generalization[24]. In this study, $\beta = 1$ was chosen to balance between the two risks (detailed in Supplementary Note 2: model parameter determination, and Supplementary Fig. 15). In order to prevent overfitting, the optimal number of training rounds was determined using additional breast cancer data. Only the model state at which breast cancer accuracy is maximized is used for inference.

### Data collection and processing
We obtained 74 human single-cell transcriptome sequencing datasets from the TISCH database[27], which includes malignant and non-malignant cells from 17 human tissues, including the bladder, blood, bone, brain, breast, colorectum, eye, head and neck, kidney, liver, lung, lymph nodes, nervous system, pancreas, pelvis, skin, and stomach. Three of the tissue datasets (bladder, kidney, lymph nodes) were discarded due to the absence of malignant cells, leaving 14 datasets for use in the subsequent steps. We used the cell annotations from the database as training labels. Following global scaling normalization ('NormalizeData' function) from Seurat[67], the values in the single-cell expression matrix of the downloaded dataset were log-transformed. To achieve a ratio of 1:1 (a sensitivity analysis on the ratio is detailed in Supplementary Fig. 16), malignant cells and non-malignant cells in each tissue were downsampled. The cells in the 14 datasets were then divided into training and validation sets in a ratio of 4:1. The variance of the normalized values across all cells was used to rank the features

(genes) and choose Highly Variable Genes (HVGs). More details on dataset collection and merging, balanced sampling, feature selection and model parameter determination are in Supplementary Note 2. After cross-validation, 328230 cells from 13 distinct tissues were finally utilized and cross-validation are detailed in Supplementary Note 2 and Supplementary Fig. 6.

### Performance evaluation

To evaluate Cancer-Finder's performance, we conducted both internal and external validation. After training the model, we determined the prediction accuracy for each of the 13 tissues from TISCH for internal validation. For external validation, we collected 10 datasets (detailed in Supplementary Tables 1, 2) and log-transformed the global scaling normalized expression matrix using SCANPY 1.9.1's '*pp.normalize_total*' and '*log1p*' functions. Classification of the malignancy of the cells was based on the annotation of the relevant initial research. Two datasets (datasets 1 and 2) served as gold standard datasets, with the remaining eight datasets serving as silver standard.

### Comparison with other methods

Cancer-Finder, CaSee[14], CopyKAT[13], SCEVAN[35] and ikarus[17] were evaluated on the 10 external validation datasets via the default parameters for fairness. CopyKAT, SCEVAN and ikarus were evaluated on a CPU platform, whereas CaSee and Cancer-Finder were examined on a GPU platform. For each dataset, we applied the pre-processing method suggested by each tool (e.g., normalization) and then used the processed dataset as the input for each tool. We recorded the total time between reading the input data and producing the output. To evaluate the performance of the algorithm on each dataset, true positives (TP), true negatives (TN), and accuracy (or similarity) were computed. Here, the accuracy (or similarity) was calculated as follows:

$$\text{ratio}_i = \frac{\text{TP}_i + \text{TN}_i}{N_i} \qquad (6)$$

$\text{TP}_i, \text{TN}_i$ and $\text{ratio}_i$ represent the true positives, true negatives and accuracy on dataset i, and $N_i$ corresponds to the number of cells in the dataset. For $n$ datasets, the average accuracy (or similarity) is calculated as:

$$\text{ratio}_{\text{weighted}} = \frac{\sum_{i=i}^{n}(\text{ratio}_i{}^*N_i)}{\sum_{i=i}^{n}N_i} \qquad (7)$$

**Test of CaSee**. CaSee was obtained from GitHub (https://github.com/yuansh3354/CaSee). The default yaml file (batch_size: 128, max_epochs: 20, lr: 0.0005) was employed. In order to make a fair comparison, CaSee was run without any prior knowledge (the organization type and available marker were set to "Unknown" and "null"). CaSee utilizes transductive learning, and we conducted training and inference five times using the default reference bulk data.

**Test of CopyKAT**. CopyKAT(V1.1.0) was obtained from GitHub (https://github.com/navinlabcode/copykat). The default parameters provided by GitHub were used (id.type = "S", ngene.chr = 5, win.size = 25, KS.cut = 0.1, distance = "euclidean", output.seg = "FLASE", plot.genes = "TRUE", genome = "hg20"). The input was the raw count 'csv' file specified by the documentation. Although we anticipated the expected "aneuploid" and "diploid" results, we discovered many "not.defined" results in the CopyKAT output. The accuracy calculation did not include cells with "not.defined" predictions, with "aneuploid" considered a positive result and "diploid" a negative result. According to the CopyKAT tool's documentation, it is recommended to process samples in batches if the test dataset

contains more than 10,000 cells; therefore, we divided test data into batches based on donor groups or batches.

**Test of SCEVAN**. SCEVAN was obtained from GitHub (https://github.com/AntonioDeFalco/SCEVAN). According to the published tutorial documentation, we input the raw count expression matrix and run it through 'pipelineCNA(count_mtx)' command without any change of settings and parameters.

**Test of ikarus**. Ikarus was obtained from GitHub (https://github.com/BIMSBbioinfo/ikarus). We followed the tutorial file (tutorial.ipynb) for training and prediction. To complete the evaluation and compare fairness, we additionally retrained ikarus five times using the same strategy as Cancer-Finder (detailed in Supplementary Note 2) and independently calculated the accuracy of the retrained models.

### Training and inference on spatial data

To train and infer on 10x Visium data, the single-cell transcriptome data is simply replaced with the spatial transcriptome's spot expression matrix. No further parameter adjustment is necessary. We assigned a spot label of 1 to the spatially annotated malignant areas, while non-malignant areas are labeled as 0. Here, a Visium slide from hepatocellular carcinoma (HCC-4L) was used to determine the number of training epochs, playing the same role as breast cancer data in the single-cell dataset.

For other types of ST data, models pre-trained on single cell data are used to annotate MERFISH[46] and Slide-seq[47] data, and a model pre-trained on 10x Visium data is used to annotate legacy ST[48] data. Additional details are available in Supplementary Fig. 7 and Supplementary Note 5.

### Analyzing ccRCC ST and TCGA datasets

Single cell reference and 10 ccRCC Visium slides containing tumor-normal interface were initially downloaded from a previously published study[54]. According to the original study, three of them were technical duplicates and two of them failed to pass the quality control and were thus excluded from subsequent analysis. Mitochondrial genes were also filtered out from the result of interpretability module. Spatially informed cell proportion on ST spots was calculated via CARD algorithm[19]. And 530 ccRCC patients with both RNA-seq data and survival information were extracted from the TCGA project. For ST data and bulk TCGA datasets, the 10-gene signature score was annotated utilizing single sample Gene Set Enrichment Analysis (ssGSEA)[57]. Spots containing non-zero expression of at least 9 genes were defined as the co-localization state. The median value of the signature score was used to group ccRCC cohorts, and the Kaplan-Meier method was applied for survival analysis.

### Software and hardware information

All training and testing on Cancer-Finder are performed on the GPU platform: CPU, Intel® Xeon® Silver 4216 CPU @ 2.10 GHz; GPU, NVIDIA GeForce RTX 3090; Memory: 512 G Bytes. Software versions are Ubuntu 18.04, Python 3.9.16, Pytorch version 1.13.1 + CUDA version 11.6. For CaSee, the tests were also performed on the GPU platform. For ikarus, CopyKAT and SCEVAN, since no GPU is used, we tested on an additional CPU platform in order to save resources: CPU, Intel® Xeon® Gold 6242 CPU @ 2.80 GHz; Memory: 1 T Bytes.

### Reporting summary

Further information on research design is available in the Nature Portfolio Reporting Summary linked to this article.

## Data availability

All data used in this study were previously published and publicly available. The training and validation data used in this study are

available in the TISCH[27] database (http://tisch1.comp-genomics.org/). Most external validation single-cell data used in this study are available in the Gene Expression Omnibus (GEO) under accession codes: cell line[28] data, GSM3618014; medulloblastoma[29] data, GSE155446; hepatoblastoma[32] data, GSE180665; head and neck cancer[33] data, GSE180268; breast cancer[13] data, GSE148673; circulating tumor cells[30,31] data, GSE109761. The remaining single-cell data for external validation, derived from a prior pancreatic cancer study[34], are accessible on the website (https://lambrechtslab.sites.vib.be/en/pan-cancer-blueprint-tumour-microenvironment-0). Please note that accessing this pancreatic cancer data requires registration. Peripheral blood mononuclear cell data from healthy donors used in this study are available on the website of 10x Genomics (https://www.10xgenomics.com/resources/datasets/10-k-peripheral-blood-mononuclear-cells-pbm-cs-from-a-healthy-donor-single-indexed-3-1-standard-4-0-0). Please note that accessing these data from 10x Genomics website requires registration. For spatial transcriptome data, colorectal cancer[43] data used in this study are available in the Genome Sequence Archive (GSA) under accession code HRA000979, hepatocellular carcinoma and intrahepatic cholangiocarcinoma[42] data used in this study are available in the GSA under accession code HRA000437. Please note that access to these GSA data is restricted, and requests for access can be made through the GSA access committee. 10x Visium spatial transcriptome data for breast and ovarian cancer used in this study are available on the 10x Genomics website (BRCA1, https://www.10xgenomics.com/resources/datasets/human-breast-cancer-block-a-section-1-1-standard-1-1-0; BRCA2: https://www.10xgenomics.com/resources/datasets/invasive-ductal-carcinoma-stained-with-fluorescent-cd-3-antibody-1-standard-1-2-0; and ovarian cancer data, https://www.10xgenomics.com/resources/datasets/human-ovarian-cancer-whole-transcriptome-analysis-stains-dapi-anti-pan-ck-anti-cd-45-1-standard-1-2-0). Please note that accessing these data from 10x Genomics website requires registration. 10x Visium ST data from renal cell carcinoma[44] (for training and identifying the gene signature) used in this study are available in the GEO under the accession code GSE175540. Renal cell carcinoma data with tumor-normal interface used in this study are available on the website https://data.mendeley.com/datasets/g67bkbnhhg/1. Data from other ST technologies (Slide-seq[49], legacy ST[50], and MERFISH) used in this study are available in the GEO under accession code GSE200278, GSE144239 and the website of MERSCOPE (https://info.vizgen.com/ffpe-showcase), respectively. Please note that accessing these MERSCOPE data requires registration. Source data are provided with this paper.

## Code availability

The code is available under the MIT license at https://github.com/Patchouli-M/SequencingCancerFinder and archived at https://doi.org/10.5281/zenodo.10505736[73].

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

## Acknowledgements

This work was supported by National Natural Science Foundation of China (NSFC) 22104080 and National Key R&D Program of China (2022YFA1104200) to Ji.S., NSFC 21735004 and 21927806 to C.Ya., and

Innovative research team of high-level local universities in Shanghai SHSMU-ZLCX20212601.

## Author contributions

Conceptualization, J.S.; Investigation, Z.Z., J.H., Z.Y., F.L.,L.D., and J.Y.; Writing, J.S., C.Y., Z.Z., Z.Y. and J.H.; Supervision, J.S., C.Y., G.O., J.Z.and T.W.

## Competing interests

The authors declare no competing interests.
