## [Peer Review File · Nature Communications]

REVIEWER COMMENTS

Reviewer #1 (Remarks to the Author): Expert in single-cell RNA-seq and spatial transcriptomics, cell identification method development, cancer genomics and tumour microenvironment

In their manuscript “Cancer-Finder: Domain generalization enables general cancer cell annotation in single-cell and spatial transcriptomics” Zhong, Hou, and Yao et al. present a new tool, Cancer-Finder, for the identification of malignant cells/spots in single-cell and spatial transcriptomics data. Cancer-Finder consists of a domain generalization-based deep-learning algorithm that is trained on single cell RNAseq data or spatial transcriptomics data in order to learn general features of malignant cells across cancer types and enable generalizable identification of malignant cells across cancers. Users can either use the pre-trained models or train their own models. The code is provided on github. In principle this is a nice and useful tool for the cancer research community especially for pan-cancer scRNAseq and spatial transcriptomics studies and will, if its promises hold true, significantly improve and standardize the annotation of malignant cells, which is indeed an important task in cancer studies. The work is well presented overall.

Major comments:

1. To really solidify the claim of generalizability across cancer types a more thorough and systematic “external validation” should be done: specifically, training should be performed on all cancer types but one and then tested on then one cancer type that had not been used for training. Across the wide range of cancer types used in this study, this procedure would give a realistic impression of what to expect from the model if a researcher wants to use it for a completely unseen tumor type. The authors already do this in a very limited form (for two cancer types) for the spatial transcriptomics data.
2. The performance is currently quantified by accuracy and similarity (two names for the same thing $(TP+TN/N)$ where accuracy is used for the external validation set and similarity for the internal validation). It might be good to make this distinction clearer and why it is necessary to use two terms for supposedly the same thing. If it is not the same thing then the respective section in the methods would need to be adjusted.
3. Only using accuracy as performance measure is prone to be biased by class imbalance. Either the classes (cancer/non-cancer) have to be balanced or measures should be used that are less prone to be biased by class imbalance like ROC-AUC or even better AUPRC. Precision/recall-F1 score would also an alternative.

4. Fig 3 panel a, the pie-charts don't correspond to what is displayed in the bar-charts. And why does head and neck not have any cancer cells?

Minor comments:

1. There are several spelling mistakes especially in the figures. Thorough spellchecking is recommended. E.g. Fig 1: Expression *prefile*, Fig. 2h: Head and *neak*

2. A bit more discussion about the logic behind the presented concept might be warranted: If, as the authors argue in the introduction, current approaches to identify malignant cells are not good then how can their model that is trained based on those annotations can be good. Is the rationale that across a lot of datasets and cancer types the model can learn beyond these (likely non-consistent mistakes)? In this case is it still appropriate to use the existent annotations as ground truth given that the model could actually be better? To work out this "conundrum" an assessment of the misclassified cells/spots would be necessary. This workup might be beyond the scope of this work but could potentially reveal further strengths of the approach.

Reviewer #2 (Remarks to the Author): Expert in single-cell RNA-seq and spatial transcriptomics analysis, bioinformatics, cancer genomics, and deep learning

The authors developed a domain generalization-based deep learning algorithm designed to efficiently and accurately detect malignant cells and regions in spatial transcriptomics. The research objective is well-defined, and the authors successfully demonstrate the Cancer-Finder's proficiency in identifying malignant cells and regions concisely and intuitively. The potential appeal of this study to pan-cancer researchers adds significance to the work. The manuscript is also well written and easy to follow. However, there are several aspects that require improvement before it can be considered for publication.

Major concerns:

1. The authors should provide additional evidence to support the superiority of Cancer-Finder in identifying malignant cells and regions. Notably, they should consider comparing Cancer-Finder with SCEVAN, a tool published in Nature Communications, which demonstrates automatic and accurate discrimination of malignant and non-malignant cells. SCEVAN outperforms CopyKAT in terms of accuracy and speed in identifying malignant cells and can also detect the clonal copy number substructure of tumors from single-cell data. Therefore, including a comparison with SCEVAN would strengthen the study.

2. The ability of Cancer-Finder to analyze other types of spatial transcriptomic (ST) data, such as image-based techniques like MERFISH and STARmap, should be systematically evaluated. While the current model mainly utilizes sequencing-based ST data like 10X visium, exploring image-based ST data will add value to the research. The authors should consider expanding the complexity of the training and test datasets to benchmark Cancer-Finder's performance comprehensively.
3. The authors reported an accuracy of over 90% for malignant cell identification. To improve transparency, they should discuss the reasons behind the mispredictions that led to cells being incorrectly classified.
4. In Figure 4b, the authors only consider OV and ICC as test datasets when the test cancer type is not included in the training data. It would be beneficial to evaluate the results using all available datasets as the test set.
5. The memory usage and computational speed of Cancer-Finder should be further compared to other tools, especially under different cell quantities, including large datasets with over a million cells. This information will provide insights into the tool's performance when dealing with substantial data volumes.
6. In Figure 5c, the predicted cancerous spots by Cancer-Finder exhibit a more dispersed distribution compared to the pathological annotations. Will Cancer-Finder perform better if the authors incorporate spatial location information into the model.
7. The methodological description of Cancer-Finder, particularly the introduction of the model in Figure 1, requires further clarification. Important concepts like "domains" need to be clearly defined, and the algorithmic foundation leading to Cancer-Finder's higher accuracy and speed compared to other tools should be elucidated.
8. The authors should demonstrate the potential application of Cancer-Finder more effectively. The current application of Cancer-Finder to clear cell renal cell carcinoma (ccRCC) in Figure 5 seems unrelated to showcasing its performance. Instead, applying Cancer-Finder to larger datasets to highlight its computational speed and accuracy would be more relevant.
9. Some tumor samples exhibit obvious pre-cancerous states in addition to normal and cancerous regions, which attract significant attention from the researchers in the field. Can the author expand the application scope of Cancer-Finder to identify pre-cancerous states in spatial transcriptomics?
10. To generate broader interest in their application, the authors are encouraged to utilize more publicly available data to establish pre-trained models for pan-cancer single-cell transcriptomics and spatial transcriptomics.
11. To improve the usability of the tool, the authors should enhance the readability of the tutorial and code. Specific suggestions include providing more detailed tutorials, describing the input data format and output, offering additional options for input data formats (e.g., anndata format), and carefully reviewing the source code of Cancer-Finder to identify and rectify any errors.
12. There are some errors in the code file, such as the help description of parameters "label_str" being the same as "gene_num" in the "get_args" function of the "args_utils.py" file. The authors should review the code thoroughly to correct such discrepancies.

Minor concerns:

1. The color differences corresponding to the tissues in figure 2a are too small, making it difficult to distinguish the tissue types in figure 2b and figure 2g.
2. Please comment on the worse performance of bone dataset in Figure 2h.
3. The dataset numbers (Line 244) should be provided alongside the corresponding dataset names in figure 3a and 3b.

Reviewer #3 (Remarks to the Author): Expert in cancer genomics, renal cancers, single-cell RNA-seq, spatial transcriptomics, and tumour microenvironment

Zhong et al present Cancer-Finder: Domain generalization enables general cancer cell annotation in single-cell and spatial transcriptomics. In general, the tool could be of use for the cancer genomic community, and they have assessed the tool with a large number of single cell and spatial datasets. The ability of a tool to discern cancer cells, and those of the stroma and immune system from single cell and ST data, would be useful.

However, I have concerns about the evidence provided. This principally centres around nomenclature and the ability of the tool to determine cell state when there are admixed normal cells. The ST data presented, especially that for ccRCC data, does not have any non-tumour regions, and for these to be annotated as such is misleading. There will be immune infiltrates that could be immune niches, but importantly they are part of the overall tumour structure. The authors need to test the model on systems that either have infiltrative cancer types, or slides that traverse the tumour normal interface. The authors might have already done this, but due to the poor annotation of the slides provided in the manuscript, it is impossible to assess.

Lines 42-43: The poor correlation is disputed, particularly in the light of IO agents, and regardless is not supported by data in this study. Having this statement at the end of the abstract is misleading as it infers that this is a conclusion you have drawn from your data.

Lines 44-45: I am not convinced that you have discovered any new biological insights. Please justify or remove.

Lines 59-60: You have missed the ST deconvolution methods that use a reference to determine the cellular makeup of tissue such as cell2location.

Lines 97-99: The ST slides you have tested on are all cancerous, ie there is an admixture of cancer cells, immune cells, and stromal cells. Measuring the effectiveness of the tool between these types is a rather low barrier. I would suggest that a better comparison would be to test the model using samples that include the normal cells that make up the cell of origin for the cancer cells. This is because the cell of origin may be the cells most transcriptionally similar to the cancer. The test set could be samples where the cancerous tissue is admixed with normal epithelial cells, such as prostate cancer <https://www.nature.com/articles/s41586-022-05023-2>, or samples where the tumour normal interface was included [https://www.cell.com/cancer-cell/fulltext/S1535-6108\(22\)00548-7](https://www.cell.com/cancer-cell/fulltext/S1535-6108(22)00548-7). This is a particular concern as your training set does not contain either benign or malignant cells of these tissue types. Without this comparison, the performance of your method cannot be assessed.

Lines 159-163: As before, separating stromal and immune cells from malignant cells is perhaps the most straightforward classification. However, this plot does not address the question of classifying cancer cells from their normal cell of origin.

Lines 261-262: The plots are poorly labelled and the colours not annotated. It should be clear what the pathologists have labelled, especially in the 'normal' areas - do these have normal architecture or are they simply regions of tumour that have high stromal or immune infiltration?

Lines 282-283: This association is not clear, and there is conflicting evidence in the literature.

Line 290: This is misleading. The entire biopsy shown is tumour, but some areas have higher immune infiltration. It is all tumour region!

Lines 311: again, these are not "non-tumour" regions.

Reviewer #4 (Remarks to the Author): Expert in single-cell RNA-seq and spatial transcriptomics methods and analysis, bioinformatics, and deep learning; co-reviewed with Reviewer #5

Summary: Zhong and colleagues introduce "Cancer-Finder," a domain generalization-based deep-learning algorithm that identifies cancer cells from single-cell data with an average accuracy of 91.60%. The authors claim that cancer-finder achieves high accuracy and generalizes well across different tissues and cancer types. The authors then attempt to apply Cancer-Finder on 23 ccRCC ST and observe evidence that shows gene expression patterns related to inflammation and CD8+ T cell exhaustion in cancerous spots, providing valuable insights for cancer research and diagnosis.

The paper's topic is crucial, showing promising results, but lacks sufficient evidence for Cancer-Finder's generalization. Rigorous cross-validation using leave-datasets-out methodology, detailed data preprocessing explanations, and interpretability efforts are necessary for credibility. Comprehensive evaluation with various metrics, including sensitivity analysis, is crucial. Without addressing these issues, the study's claim of a generalized model is underpowered, hindering its impact on the cancer annotation community.

Major Critiques:

1- Spatial Transcriptome Data Acquisition Technology: Spatial transcriptome data acquisition platforms, like Spatial Transcriptomics, Slide-seq, Visium, MERFISH, and STARmap, have specific strengths and limitations impacting data quality, resolution, and sensitivity. Rigorous evaluation with multiple datasets and cross-validation is crucial to assess model robustness and identify potential biases from platform variability. The paper should highlight the used platform, address biases, and demonstrate the impact of domain adaptation and transfer learning on generalization. Developing the model on only one platform risks lacking “generalizability”.

2- Cross-validation Techniques: The paper introduces important findings, but there are concerns regarding the cross-validation techniques and the “generalizability” of the model. While the authors mention using five-fold cross-validation, there is a lack of detailed information on the process. It is essential to clarify whether they used leave-cells-out or leave-datasets-out cross-validation, as the former can lead to overestimating generalizability by “memorizing” unique signatures of individual datasets. To strengthen the paper's claims, I suggest repeating the analysis with leave-datasets-out cross-validation, as it can better assess domain generalization capabilities and highlight the model's performance across different cancer types. Addressing these issues would provide a more comprehensive understanding of the model's true generalizability and improve the study's impact in the field of cancer annotation. To rigorously assess the model's performance and domain generalization capabilities, the authors must conduct experiments using leave-datasets-out cross-validation. Additionally, investigating leave-one-cancer-type-out cross-validation would be essential, as it directly relates to the paper's claims of generalizability. Without addressing these issues, the study's credibility is at risk, and its claims about model generalization in the cancer annotation community lack solid evidence.

3- Data Preprocessing: The paper briefly mentions data preprocessing steps but lacks in-depth explanations and the rationale behind these choices. Elaborating on the data preprocessing methods and their potential impact on results is crucial for better understanding the model's performance. The paper must provide a comprehensive and detailed explanation of the data preprocessing methods used, along with the rationale behind each choice. Understanding the impact of preprocessing on results is crucial for interpreting the model's performance accurately.

4- Interpretability of the Model: The paper acknowledges that the deep learning model can be hard to interpret. While this is expected, it is needed that the authors attempt to shed some light on the model's inner workings using techniques like saliency maps. This effort can enhance the credibility of the results and aid in clinical applications. Despite acknowledging the model's black box nature, the authors should make an effort to shed light on the model's inner workings using techniques like saliency maps. This will enhance the credibility of the results and enable potential clinical applications.

5- Evaluation Metrics: F1 score and other metrics, such as ROC curves, are not discussed. It would be valuable to include a comprehensive evaluation of the model's performance using various metrics to provide a more comprehensive assessment. The paper should include a thorough evaluation of the model's performance using various metrics, including F1 score and ROC curves. This comprehensive assessment will provide a more robust understanding of the model's strengths and weaknesses.

6- Bias and Sensitivity Analysis: The use of a 1-to-1 ratio in cancer cell classification is commendable to prevent bias towards specific cell types. However, the paper lacks sensitivity analysis, especially for spatial data. Conducting simulations and providing insights into the model's sensitivity to different input dimensions would strengthen the paper's conclusions. The use of a 1-to-1 ratio for cancer cell classification is commendable, but the paper must conduct sensitivity analysis, especially for spatial data. Simulations should be performed to gain insights into the model's sensitivity to different input dimensions, ensuring the model's reliability and generalizability.

7- Ambiguous Feature Extraction Module: The description of the feature extraction module is ambiguous and lacks clarity. The number of neurons in each layer and the activation functions used in the fully connected layers should be clearly specified. Additionally, details about how the dropout probability was determined to prevent overfitting need to be provided.

8- Incomplete Training Approaches: The section on training approaches for domain generalization tasks lacks complete information. The equation provided for the loss function needs further elaboration, and the rationale behind using the proposed risk extrapolation method should be explained in detail. Additionally, the role of the hyperparameter β and how it affects the training process should be clarified.

9- Data Collection and Processing: The description of data collection and processing is inadequate. The TISCH database is mentioned, but there is no information on the specific datasets used and their characteristics. Additionally, details on how the down-sampling of malignant and non-malignant cells was performed to achieve a 1:1 ratio need to be provided.

10- Provided codes: The provided code seems relatively simple and concise. While simple and concise, the absence of comments or explanatory notes is a major flaw, hindering usability and maintainability. Thorough comments must be added to improve the code's quality and accessibility.

Minor Critiques:

Line 154: There is missing space after "." and "The" ((2a).The).

Line 272: What type of "general predictions" do the authors mean?

Line 293: For Figures 5c and 5d, the figure's scale is unclear to me.

Line 382, Please delete (<3000words) from in front of the Methods.

Reviewer #5 (Remarks to the Author): Expert in single-cell RNA-seq and spatial transcriptomics methods and analysis, bioinformatics, and deep learning; co-reviewed with Reviewer #4

I thoroughly co-reviewed this manuscript with one of the reviewers who provided the listed reports as part of the Nature Communications initiative to facilitate training in peer review and appropriate recognition for co-reviewers.

Manuscript ID: NCOMMS-23-25528

REVIEWER COMMENTS

Reviewer #1 (Remarks to the Author): Expert in single-cell RNA-seq and spatial transcriptomics, cell identification method development, cancer genomics and tumour microenvironment

In their manuscript “Cancer-Finder: Domain generalization enables general cancer cell annotation in single-cell and spatial transcriptomics” Zhong, Hou, and Yao et al. present a new tool, Cancer-Finder, for the identification of malignant cells/spots in single-cell and spatial transcriptomics data. Cancer-Finder consists of a domain generalization-based deep-learning algorithm that is trained on single cell RNAseq data or spatial transcriptomics data in order to learn general features of malignant cells across cancer types and enable generalizable identification of malignant cells across cancers. Users can either use the pre-trained models or train their own models. The code is provided on github. In principle this is a nice and useful tool for the cancer research community especially for pan-cancer scRNAseq and spatial transcriptomics studies and will, if its promises hold true, significantly improve and standardize the annotation of malignant cells, which is indeed an important task in cancer studies. The work is well presented overall.

Authors' response: Sincere thanks to the reviewer for the positive feedback and insightful suggestions. We have considered the comments carefully and revised the manuscript accordingly. Please see our detailed response provided below.

Major comments:

1. To really solidify the claim of generalizability across cancer types a more thorough and systematic “external validation” should be done: specifically, training should be performed on all cancer types but one and then tested on then one cancer type that had not been used for training. Across the wide range of cancer types used in this study, this procedure would give a realistic impression of what to expect from the model if a researcher wants to use it for a completely unseen tumor type. The authors already do this in a very limited form (for two cancer types) for the spatial transcriptomics data.

Authors' response: We appreciate the reviewer's comment, which can make our paper more solid. As shown in the revised **Supplementary Figure 6**, we excluded the data for each type of cancer from our training set, trained with the remaining data, and then used the trained model to make predictions on the excluded data. Across 14 tissues,

Cancer-Finder demonstrated an overall average accuracy (weighted accuracy) of 88.54%. This result has been added to the revised manuscript (lines 277-279 page 9) and revised **Supplementary Figure 6** in SI.

Notably, Cancer-Finder performed poorly (accuracy < 0.8) on a few datasets, which primarily consisted of hematologic tumors (including AML, ALL, MM and AEL). The poor performance on hematologic tumors is typically attributable to the fundamentally distinct mechanisms that underlie hematologic and solid tumors. Therefore, we added a note to the revised manuscript (lines 437-441 page 13) stating that this portion of the pre-trained model is not recommended for hematologic tumor cell identification.

For the solid tumor with lowest accuracy (STAD), we compared the database annotation results to the original study (*Cell Rep.*, 2019, 27(6), 1934–1947) and found significant discrepancies. Detailly, nine patients with Non-Atrophic Gastritis (NAG), Chronic Atrophic Gastritis (CAG), Intestinal Metaplasia (IM), or Early Gastric Cancer (EGC) provided samples for the STAD dataset (GSE134520). According to the annotations of the TISCH database, malignant cells are present in the samples of patients with NAG, CAG, and IM diseases, but not in the samples of patients with early gastric cancer (EGC). However, according to the original study, patients with NAG and CAG should not have malignant cells in their samples, whereas patients with EGC should have many malignant cells in their samples. This discrepancy is likely the result of errors introduced by the database during the data collection process. This merely demonstrates that our tool can be a useful aid for relevant database annotation and error detection. All of these results and discussions have been added to the legend of revised **Supplementary Figure 6**.

For spatial transcriptomics data, we have collected ST expression matrices and corresponding labels from original articles (mostly manually annotated) that cover a total of 14 slides from 6 distinct cancer types. Based on this data, we refined **Figure 4b** to provide a more comprehensive characterization of Cancer-Finder. The overall performance of Cancer-Finder is relatively good, but the accuracy of some untrained cancers is less desirable. This may be due to the fact that the overall training dataset is still limited, as we are unable to collect truthful labels for the majority of available ST data. Compared to ST, the dataset for single cells is larger. Therefore, its training results can yield better performance, even for cancer types that were not included in the training data (revised **Figure 4b**). As data accumulates, we will collect additional data for future training and updates. This discussion has been added in lines 314-318 page 10-11 in this revision.

Revised Supplementary Figure 6. Results of leave-one-cancer-type-out and leave-datasets-out cross-validations. a,b Results of leave-one-cancer-type-out cross-validation on single-cell data and balanced single-cell data. Leave-one-cancer-type-out cross-validation was performed by excluding scRNA-seq data from one cancer, training Cancer-Finder with data from other cancers, and predicting cell annotation labels for the excluded cancer. Cancer-Finder performs well (accuracy > 0.8) on most cancers, but its performance is limited on hematologic tumors (colored in grey), possibly due to the significant difference between hematologic and solid tumors. Notably, a discrepancy was discovered between the original study (*Cell Rep*, 2019, 27(6), 1934–1947) and TISCH's annotation on cancer with the lowest accuracy (colored red), which may have been caused by database collection errors in the database. c,d Results of leave-datasets-out cross-validation on single-cell data and balanced single-cell data. Leave-datasets-out cross-validation was conducted by excluding one scRNA-seq dataset at a time, training Cancer-Finder with data from other datasets, and predicting cell annotation labels for the excluded dataset. On four datasets (colored red), discrepancies were discovered between the original study (*Science*, 2020, 367(6476):405-411, *Cell Rep.*, 2019, 27(6), 1934–1947, *Cancer Discov.*, 2019,9(12):1708-1719, *Nat. Commun.*, 2020, 11(1):2285) and TISCH's annotation, possibly due to database collection errors. These results demonstrated that Cancer-Finder can be a useful aid for relevant database annotation and error detection. The datasets represented by the green bars consist of either all-malignant or all-non-malignant cells and are therefore not included in the balanced single-cell data.

Training type	Training set	Test set	Similarity
The test cancer type is included in the training data	3 HCC, 3 CRC, 1 BRCA, 1 OV, 3 RCC, 1 ICC	BRCA	97.37%
	2 HCC, 3 CRC, 2 BRCA, 1 OV, 3 RCC, 1 ICC	HCC	95.54%
	3 HCC, 2 CRC, 2 BRCA, 1 OV, 3 RCC, 1 ICC	CRC	89.80%
	3 HCC, 3 CRC, 2 BRCA, 1 OV, 2 RCC, 1 ICC	RCC	82.00%
The test cancer type is not included in the training data	3 HCC, 3 CRC, 1 OV, 3 RCC, 1 ICC	BRCA	95.02%
	3 CRC, 2 BRCA, 1 OV, 3 RCC, 1 ICC	HCC	87.50%
	3 HCC, 2 BRCA, 1 OV, 3 RCC, 1 ICC	CRC	88.57%
	3 HCC, 3 CRC, 2 BRCA, 3 RCC, 1 ICC	OV	86.37%
	3 HCC, 3 CRC, 2 BRCA, 1 OV, 1 ICC	RCC	61.62%
	3 HCC, 3 CRC, 1 BRCA, 1 OV, 1 RC	ICC	58.95%

Revised Figure 4b. Specifics regarding the datasets used during spatial transcriptome training and testing, as well as the accuracies obtained.

2. The performance is currently quantified by accuracy and similarity (two names for the same thing (TP+TN/N) where accuracy is used for the external validation set and similarity for the internal validation). It might be good to make this distinction clearer and why it is necessary to use two terms for supposedly the same thing. If it is not the same thing then the respective section in the methods would need to be adjusted.

Authors' response: We thank the reviewer's valuable advice. We use two distinct terms for the same metric because the label reliability in the benchmark datasets differs. We use accuracy to describe the accuracy of Cancer-Finder on the gold standard dataset because the reference labels of cells on the gold standard dataset are extremely trustworthy. Because the reference labels on the silver standard dataset were annotated by other studies and may not be completely reliable, we use similarity to characterize Cancer-Finder's prediction of labels on the silver standard dataset. To prevent confusion, the specific description was added to revised **Supplementary Note 3** of SI.

3. Only using accuracy as performance measure is prone to be biased by class imbalance. Either the classes (cancer/non-cancer) have to be balanced or measures should be used that are less prone to be biased by class imbalance like ROC-AUC or even better AUPRC. Precision/recall-F1 score would also an alternative.

Authors' response: We thank the reviewer's valuable advice. In this new revision, we have added precision, recall, F1 scores, AUROC and AUPRC values to each dataset to provide a more complete picture of Cancer-Finder's performance (revised **Supplementary Figure 2-5** and **Figure 3b**).

During algorithm comparisons, we included accuracy, precision, recall, and F1 scores. We were only able to calculate the AUROC and AUPRC values for Cancer-Finder because other algorithms did not provide a continuous value (such as the Softmax value in Cancer-Finder) that could be used for these two metrics.

Revised Figure 3b. Comparison of Cancer-Finder's prediction accuracy to four other cell annotation algorithms. Cancer-Finder demonstrated significantly greater accuracy and similarity (to the labels from the original studies) than other tools across various cancers. All tests were conducted in parallel five times. The presence of an 'NA' denotes that the method returns an error and cannot be executed with these data.

Revised Supplementary Figure 2. Comparison of the Precision of Cancer-Finder and four other cell annotation algorithms. Each test was conducted five times in

parallel. The presence of an ‘NA’ denotes that the method returns an error and cannot be executed with these data, or that the dataset contains only positive or negative samples, therefore the indicator cannot be calculated.

Revised Supplementary Figure 3. Comparison of the F1 score and the Recall rate of Cancer-Finder and four other cell annotation algorithms. Each test was conducted five times in parallel. The presence of an ‘NA’ denotes that the method returns an error and cannot be executed with these data, or that the dataset contains only positive or negative samples, therefore the indicator cannot be calculated.

Revised Supplementary Figure 4. AUROC of Cancer-Finder’s prediction results across test datasets. Each test was conducted five times in parallel, and the average AUROC was presented.

Revised Supplementary Figure 5. AUPRC of Cancer-Finder’s prediction results across test datasets. Each test was conducted five times in parallel, and the average AUPRC was presented.

4. Fig 3 panel a, the pie-charts don’t correspond to what is displayed in the bar-charts. And why does head and neck not have any cancer cells?

Authors’ response: We thank the reviewer’s valuable advice. Due to the logarithmic transformation of the x-axis in the bar charts, the portion represented by blue indicating normal cells appears relatively small. To avoid confusion, we modified the left bar to display only the total number of cells; the ratio of cells is only displayed on the right pie chart.

According to the original study (*Nature*, 2021, 597(7875):279-284), the head and neck dataset exclusively concentrated on immune cells, with cancer cells being experimentally excluded. This note has been added to the Figure Legend.

Revised Figure 3. Performance comparison with existing methods and application to a large database. a, Data structure of ten external validation datasets with varying cell counts and proportions of malignant cells. Among them, the head and neck dataset exclusively concentrated on immune cells, with malignant cells being experimentally excluded. b, Comparison of Cancer-Finder's prediction accuracy to four other cell annotation algorithms. Cancer-Finder demonstrated significantly greater accuracy and similarity (to the labels from the original studies) than other tools across various cancers. All tests were conducted in parallel five times. The presence of an 'NA' denotes that the method returns an error and cannot be executed with these data. c, Comparison of Cancer-Finder's inference speed to the other four methods. 'NA' indicates that the method could not run correctly on the data. This study utilized a maximum memory size of 512G Bytes. d, Comparison of Cancer-Finder's memory consumption to the other four methods. 'NA' indicates that the method could not run correctly on the data. e, Up-regulated genes identified from predicted malignant cells. Full names of the cancers are detailed in **Supplementary Table 15**.

Minor comments:

1. There are several spelling mistakes especially in the figures. Thorough spellchecking is recommended. E.g. Fig 1: Expression *prefile*, Fig. 2h: Head and *neak*

Authors' response: We apologize for these errors and appreciate reviewer's careful reading. We rechecked the manuscript and fixed these errors.

2. A bit more discussion about the logic behind the presented concept might be warranted: If, as the authors argue in the introduction, current approaches to identify malignant cells are not good then how can their model that is trained based on those annotations can be good. Is the rationale that across a lot of datasets and cancer types the model can learn beyond these (likely non-consistent mistakes)? In this case is it still appropriate to use the existent annotations as ground truth given that the model could actually be better? To work out this "conundrum" an assessment of the misclassified cells/spots would be necessary. This workup might be beyond the scope of this work but could potentially reveal further strengths of the approach.

Authors' response: We thank the reviewer's valuable advice.

This study was proposed on the assumption that, while not entirely accurate, good databases (such as TISCH) with labels annotated based on integrated computational and manual annotation produce mostly accurate annotation results and deep learning has the potential to learn a generally accurate rule from a mostly accurate training set. To demonstrate this, we introduced varying degrees of annotation error into the training dataset by randomly selecting and changing 5%, 10%,..., and 50% of the original labels. We retrained and evaluated the model using 5-fold cross-validation (leave-cells-out) and determined that the model's accuracy was not significantly impacted when less than 35% of labels were incorrect (**Supplementary Figure 11**). This suggests that some degree of error in Cancer-Finder's training set is acceptable.

Moreover, based on the outcome of Cancer-Finder, we efficiently identified mislabeled datasets within the database, specifically the STAD data (a discrepancy was observed between the original study and TISCH's cancer annotation) (detailed in response of comment#1). This result indicates that Cancer-Finder has the potential to learn universal patterns based on correct labels and to identify potentially incorrect labels in the training dataset, thereby assisting us in further optimizing the training set. Certainly, it is undeniable that a highly accurate new dataset would be better suited to train the model, but such an endeavor would require a lot of experiments to separate malignant and non-malignant cells, making it a highly inefficient method. Instead, the universal rules discovered by our model may help to de-optimize the annotation of these currently available databases to better serve future research.

In addition, we compared incorrectly predicted cells to correctly predicted cells and discovered that incorrectly predicted cells contained significantly lower average expression of highly variable genes and fewer detected genes (Revised **Supplementary**

Figure 12). These results suggest that incorrectly predicted cells may have a lower quality of sequencing. Here, since we can only guarantee that the label of the cell within the gold standard dataset is completely accurate, we primarily investigate this problem using the gold standard datasets. In addition, as revealed by leave-platform-out cross-validation (Revised **Supplementary Figure 10**), Cancer-Finder shows a varied accuracy on data from different sequencing platforms.

These discussions have been added to the Discussion section of the revised manuscript lines 390-406 page 13.

Revised Supplementary Figure 11. Performance of Cancer-Finder when the training set is partially incorrect. Here, we demonstrated that the overall performance of Cancer-Finder remains stable in the presence of less than 35% incorrect labels by modifying the labels in the training set to incorrect annotations and then using them to train the model. Five-fold leave-cells-out cross-validation was performed for each rate (Error bars show mean \pm standard deviation of these 5 validations).

Revised Supplementary Figure 12. Comparisons between cells that were correctly and incorrectly predicted. Two gold standard datasets were used. a, Comparison of average expression of HVGs in correctly predicted and incorrectly predicted cells in the cell line dataset. Here, we use the top 2000 HVGs according to expression variance.

b, Comparison of average expression of HVGs in correctly predicted and incorrectly predicted cells in PBMC dataset. Here, we use the top 2000 HVGs according to expression variance. c, Comparison of number of detected genes in correctly predicted and incorrectly predicted cells. Paired t-tests were used. The violins are centered at median values, where the range of violins represents the interquartile range (IQR) bounded by the first quartile (Q1) and the third quartile (Q3).

Revised Supplementary Figure 10. Results of leave-platform-out cross-validations. Leave-platform-out cross-validation was performed by excluding datasets from one platform, training Cancer-Finder with data from other platforms, and predicting cell annotation labels for the excluded datasets. Each point represents the accuracy of a dataset. The dataset from mCEL-seq2 was excluded from the balanced validation because it contains only non-malignant (negative) cells. Notably, several datasets were excluded from balanced validation because they contained only all-malignant or all-nonmalignant cells. The boxes are centered at median values, where the range of boxes represents the interquartile range (IQR) bounded by the first quartile (Q1) and the third quartile (Q3).

Reviewer #2 (Remarks to the Author): Expert in single-cell RNA-seq and spatial transcriptomics analysis, bioinformatics, cancer genomics, and deep learning

The authors developed a domain generalization-based deep learning algorithm designed to efficiently and accurately detect malignant cells and regions in spatial transcriptomics. The research objective is well-defined, and the authors successfully demonstrate the Cancer-Finder's proficiency in identifying malignant cells and regions concisely and intuitively. The potential appeal of this study to pan-cancer researchers adds significance to the work. The manuscript is also well written and easy to follow. However, there are several aspects that require improvement before it can be considered for publication.

Authors' response: We are grateful for the reviewer's thoughtful and positive comments. We found the reviewers' comments to be extremely perceptive and instructive. After studying the comments, we made some changes to the manuscript. Please see our point-to-point response below.

Major concerns:

1. The authors should provide additional evidence to support the superiority of Cancer-Finder in identifying malignant cells and regions. Notably, they should consider comparing Cancer-Finder with SCEVAN, a tool published in Nature Communications, which demonstrates automatic and accurate discrimination of malignant and non-malignant cells. SCEVAN outperforms CopyKAT in terms of accuracy and speed in identifying malignant cells and can also detect the clonal copy number substructure of tumors from single-cell data. Therefore, including a comparison with SCEVAN would strengthen the study.

Authors' response: We appreciate the reviewer's comment, which can make our paper more solid. In this revision, the SCEVAN algorithm is included in the section where our method is compared to others. According to the evaluation results, we learned that Cancer-Finder is superior to SCEVAN at identifying malignant cells. The results are shown in the revised **Figure 3b**.

Revised Figure 3b. Comparison of Cancer-Finder's prediction accuracy to four other cell annotation algorithms. Cancer-Finder demonstrated significantly greater accuracy and similarity (to the labels from the original studies) than other tools across various cancers. All tests were conducted in parallel five times. The presence of an 'NA' denotes that the method returns an error and cannot be executed with these data.

2. The ability of Cancer-Finder to analyze other types of spatial transcriptomic (ST) data, such as image-based techniques like MERFISH and STARmap, should be systematically evaluated. While the current model mainly utilizes sequencing-based ST data like 10X visium, exploring image-based ST data will add value to the research. The authors should consider expanding the complexity of the training and test datasets to benchmark Cancer-Finder's performance comprehensively.

Authors' response: We appreciate the reviewer's comment.

Other than the commercial platform 10X Visium, which has been utilized in a variety of applications, other platforms have fewer use cases, and even fewer data on cancer tissues. In this situation, it is challenging to collect a large enough training set (including at least 2-3 types of cancer data) to train a pre-trained model on data from multiple platforms. Consequently, this revision focuses primarily on predicting data from other platforms using the training results from the existing training set (the pre-trained models based on scRNA-seq data and 10X Visium data). Here, we primarily focus on making predictions using datasets from one imaging-based technique (MERFISH, *Science*, 2015, 348(6233):aaa6090) and two sequencing-based techniques with different resolutions, namely Slide-seq (*Science*, 2019, 363(6434):1463-1467) and legacy ST (*Science*, 2016, 353(6294):78-82). Detailly, four MERFISH slides

(<https://info.vizgen.com/ffpe-showcase>), four Slide-seq slides (*Cell*, 2022, 185(14):2591-2608) and two legacy ST slides (*Cell*, 2020, 182(6):1661-1662) were downloaded.

Considering that MERFISH data are most similar to the single-cell form, we initially trained the model with scRNA-seq data. Here, we trained the model using the single-cell sub-matrix (containing 550 genes measured in the MERFISH data), and utilized the training results to predict malignant cells in the MERFISH dataset. As shown in the revised **Supplementary Figure 7a** and **Figure 4**, in the case of using a suitable Softmax threshold, Cancer-Finder has a high degree of accuracy on the MERFISH data. Notably, we observed that Cancer-Finder may generate false positives when the pre-trained model was applied directly to MERFISH data with the default softmax threshold (threshold = 0.5) because single-cell data and MERFISH data are not identical. Based on a MERFISH slide, the ROC curve was used to determine the optimal threshold (threshold = 0.9766), Cancer-Finder was able to accurately predict MERFISH data (accuracy: 70.69–83.84 %, AUC: 0.7707–0.8969).

Similarly, we have expanded our predictions to Slide-seq data. This is a second-generation sequencing-based ST technology with near single-cell resolution (spot diameter of 10um), so we still made predictions with the pre-trained model we obtained on the scRNA-seq dataset, and the results demonstrated that Cancer-Finder performs exceptionally well on the majority of the datasets (revised **Supplementary Figure 7b**).

Lastly, we attempted to extend the model to legacy ST slides with a larger spot (spot diameter of 100um) and made predictions utilizing a pre-trained model trained on 10x Visium slides. As shown in revised **Supplementary Figure 7c**, the performance of Cancer-Finder varies across datasets (slide 1: accuracy=0.8050, AUC=0.8227; slide 2: accuracy=0.5765, AUC=0.5650).

In conclusion, it is difficult for us to provide a good pre-trained model for every type of ST data given the current state of data accumulation. Notwithstanding, the prediction results of models trained on existing scRNA-seq and 10x Visium ST slides on sequencing data from platforms with comparable resolution are reasonably accurate. In the future, we will attempt to collect and accumulate additional data for such an exercise. We have added these discussions to the revised Result (lines 319-332, page 11), Discussion (lines 424-427 page 14), and SI (revised **Supplementary Note 5**, **Supplementary Figure 7**) parts.

The generalization capability of Cancer-Finder was also evaluated on single-cell data by removing data from a specific technique from the training set and making predictions on this data. Cancer-Finder is able to generalize effectively across techniques (with an average accuracy of 86.35%), but its accuracy fluctuates among different techniques. We have incorporated this discussion into the Revised Discussion (lines 395-398, page 13), and SI (**Supplementary Figure 10**) sections, aiming to assist users in making informed decisions.

Revised Figure 4. Expansion of Cancer-Finder to spatial transcriptomic annotation. a, Expansion of the training process to include ST data. Annotations from pathologists were used to determine which spots were malignant. Training is completed by replacing the single-cell matrix with the spot expression matrix directly. b, Specifics regarding the datasets used during spatial transcriptome training and testing, as well as the accuracy rates obtained. c, Comparison of model predictions and annotation labels

(left) from pathologists for cases where the tested cancer type was included (middle) or excluded (right) from the training data. d, Comparison between model predictions and expression of malignant marker genes on MERFISH data (up), and comparison between model predictions and annotations from original research on Slide-seq data (down).

Revised Supplementary Figure 7. Application expansion of Cancer-Finder. a, Performance of Cancer-Finder on MERFISH data. Here, the optimal Softmax threshold (threshold = 0.9766) was determined according to the ROC curve based on an external MERFISH slide. ‘*’ denotes the slide used to determine the threshold value. b, Performance of Cancer-Finder on slide-seq data. c, Performance of Cancer-Finder on ‘‘ST’’ data.

Revised Supplementary Figure 10. Results of leave-platform-out cross-validations. Leave-platform-out cross-validation was performed by excluding datasets from one platform, training Cancer-Finder with data from other platforms, and predicting cell annotation labels for the excluded datasets. Each point represents the accuracy of a dataset. The dataset from mCEL-seq2 was excluded from the balanced validation because it contains only non-malignant (negative) cells. Notably, several datasets were excluded from balanced validation because they contained only all-malignant or all-nonmalignant cells. The boxes are centered at median values, where the range of boxes represents the interquartile range (IQR) bounded by the first quartile (Q1) and the third quartile (Q3).

3. The authors reported an accuracy of over 90% for malignant cell identification. To improve transparency, they should discuss the reasons behind the mispredictions that led to cells being incorrectly classified.

Authors' response: We appreciate the reviewer's comment, which can make our paper more solid. Since we can only guarantee that the label of the cell within the gold standard dataset is completely accurate, we primarily investigate this problem using the gold standard datasets.

We compared incorrectly predicted cells to correctly predicted cells and discovered that incorrectly predicted cells contained significantly lower average expression of highly variable genes and fewer detected genes (revised **Supplementary Figure 12**). These results suggest that incorrectly predicted cells may have a lower quality of sequencing. In addition, as revealed by leave-platform-out cross-validation (revised **Supplementary Figure 10**), Cancer-Finder shows a varied accuracy on data from different sequencing platforms.

This discussion has been added to the revised Discussion (lines 395-406 Page 13) and SI (revised **Supplementary Figure 12** and revised **Supplementary Figure 10**).

Revised Supplementary Figure 12. Comparisons between cells that were correctly and incorrectly predicted. Two gold standard datasets were used. a, Comparison of average expression of HVGs in correctly predicted and incorrectly predicted cells in the cell line dataset. Here, we use the top 2000 HVGs according to expression variance. b, Comparison of average expression of HVGs in correctly predicted and incorrectly predicted cells in PBMC dataset. Here, we use the top 2000 HVGs according to expression variance. c, Comparison of number of detected genes in correctly predicted and incorrectly predicted cells. Paired t-tests were used. The violins are centered at median values, where the range of violins represents the interquartile range (IQR) bounded by the first quartile (Q1) and the third quartile (Q3).

Revised Supplementary Figure 10. Results of leave-platform-out cross-validation. Leave-platform-out cross-validation was performed by excluding datasets from one platform, training Cancer-Finder with data from other platforms, and predicting cell annotation labels for the excluded datasets. Each point represents the accuracy of a dataset. The dataset from mCEL-seq2 was excluded from the balanced validation because it contains only non-malignant (negative) cells. Notably, several datasets were

excluded from balanced validation because they contained only all-malignant or all-nonmalignant cells. The boxes are centered at median values, where the range of boxes represents the interquartile range (IQR) bounded by the first quartile (Q1) and the third quartile (Q3).

4. In Figure 4b, the authors only consider OV and ICC as test datasets when the test cancer type is not included in the training data. It would be beneficial to evaluate the results using all available datasets as the test set.

Authors' response: We appreciate the reviewer's comment. To conduct a more thorough investigation, we collected as many additional datasets as possible. To date, we have collected ST expression matrices and corresponding labels from original articles (mostly manually annotated) that cover a total of 14 slides from 6 distinct cancer types. Based on these data, we performed model training after data exclusion for each cancer type and used it to predict these excluded cancer slides; the revised **Figure 4b** display the generalization performance of Cancer-Finder in a more comprehensive manner.

The overall performance of Cancer-Finder is relatively good, but the accuracy of some untrained cancers is less desirable. This may be due to the fact that the overall training dataset is still limited, as we are unable to collect truthful labels for the majority of available ST data. Compared to ST, the dataset for single cells is larger. Therefore, its training results can yield better performance, even for cancer types that were not included in the training data (revised **Supplementary Figure 6**). As data accumulates, we will collect additional data for future training and updates. This discussion has been added in lines 314-318 page 10-11 in this revision.

Training type	Training set	Test set	Similarity
The test cancer type is included in the training data	3 HCC, 3 CRC, 1 BRCA, 1 OV, 3 RCC, 1 ICC	BRCA	97.37%
	2 HCC, 3 CRC, 2 BRCA, 1 OV, 3 RCC, 1 ICC	HCC	95.54%
	3 HCC, 2 CRC, 2 BRCA, 1 OV, 3 RCC, 1 ICC	CRC	89.80%
	3 HCC, 3 CRC, 2 BRCA, 1 OV, 2 RCC, 1 ICC	RCC	82.00%
The test cancer type is not included in the training data	3 HCC, 3 CRC, 1 OV, 3 RCC, 1 ICC	BRCA	95.02%
	3 CRC, 2 BRCA, 1 OV, 3 RCC, 1 ICC	HCC	87.50%
	3 HCC, 2 BRCA, 1 OV, 3 RCC, 1 ICC	CRC	88.57%
	3 HCC, 3 CRC, 2 BRCA, 3 RCC, 1 ICC	OV	86.37%
	3 HCC, 3 CRC, 2 BRCA, 1 OV, 1 ICC	RCC	61.62%
	3 HCC, 3 CRC, 1 BRCA, 1 OV, 1 RC	ICC	58.95%

Revised Figure 4b. Specifics regarding the datasets used during spatial transcriptome training and testing, as well as the accuracies obtained.

Revised Supplementary Figure 6. Results of leave-one-cancer-type-out and leave-datasets-out cross-validations. a,b Results of leave-one-cancer-type-out cross-validation on single-cell data and balanced single-cell data. Leave-one-cancer-type-out cross-validation was performed by excluding scRNA-seq data from one cancer, training Cancer-Finder with data from other cancers, and predicting cell annotation labels for the excluded cancer. Cancer-Finder performs well (accuracy > 0.8) on most cancers, but its performance is limited on hematologic tumors (colored in grey), possibly due to the significant difference between hematologic and solid tumors. Notably, a discrepancy was discovered between the original study (*Cell Rep.*, 2019, 27(6), 1934–1947) and TISCH's annotation on cancer with the lowest accuracy (colored red), which may have been caused by database collection errors in the database. c,d Results of leave-datasets-out cross-validation on single-cell data and balanced single-cell data. Leave-datasets-out cross-validation was conducted by excluding one scRNA-seq dataset at a time, training Cancer-Finder with data from other datasets, and predicting cell annotation labels for the excluded dataset. On four datasets (colored red), discrepancies were discovered between the original study (*Science*, 2020, 367(6476):405-411, *Cell Rep.*, 2019, 27(6), 1934–1947, *Cancer Discov.*, 2019,9(12):1708-1719, *Nat. Commun.*, 2020, 11(1):2285) and TISCH's annotation, possibly due to database collection errors. These results demonstrated that Cancer-Finder can be a useful aid for relevant database annotation and error detection. The datasets represented by the green bars consist of either all-malignant or all-non-malignant cells and are therefore not included in the balanced single-cell data.

5. The memory usage and computational speed of Cancer-Finder should be further compared to other tools, especially under different cell quantities, including large datasets with over a million cells. This information will provide insights into the tool's performance when dealing with substantial data volumes.

Authors' response: We thank the reviewer's valuable advice. In this version, we have expanded the comparison to a dataset with one million cells and revised **Figure 3c**.

We tested the inference speed of Cancer-Finder on datasets of varying sizes, including 100, 1000, 10,000, 100,000 and 1,000,000 cells (detailed in Supplementary Notes 1), and found inference times of 1.39s, 2.53s, 58.65s, 3,903.41s and 38,903.92s (~11 hours), respectively, on a computer with 16 cores, 2.1Ghz CPU and NVIDIA 3090 GPU. These results demonstrate that Cancer-Finder is significantly more efficient than other algorithms (revised **Figure 3c**, detailed in Methods). In addition, in comparison to other algorithms, Cancer-Finder uses the least amount of memory (an order of magnitude fewer than competing algorithms, revised **Figure 3d** and **Supplementary Table 14**). This section has been modified in this version (lines 257-270, page 9).

Revised Figure 3. Performance comparison with existing methods and application to a large database. a, Data structure of ten external validation datasets with varying cell counts and proportions of malignant cells. Among them, the head and neck dataset exclusively concentrated on immune cells, with malignant cells being experimentally excluded. b, Comparison of Cancer-Finder's prediction accuracy to four other cell annotation algorithms. Cancer-Finder demonstrated significantly greater accuracy and similarity (to the labels from the original studies) than other tools across various cancers. All tests were conducted in parallel five times. The presence of an 'NA' denotes that the method returns an error and cannot be executed with these data. c, Comparison of Cancer-Finder's inference speed to the other four methods. 'NA' indicates that the method could not run correctly on the data. This study utilized a maximum memory size of 512G Bytes. d, Comparison of Cancer-Finder's memory consumption to the other four methods. 'NA' indicates that the method could not run correctly on the data. e, Up-regulated genes identified from predicted malignant cells. Full names of the cancers are detailed in **Supplementary Table 15**.

Supplementary Table 14. Memory consumption comparison of five algorithms

Number of cells Algorithms	100	1,000	10,000	100,000	1,000,000
Cancer-Finder	13,532 KB	13,568 KB	3,924,116 KB	13,956,016 KB	14,865,940 KB
SCEVAN	1,657,160 KB	3,751,188 KB	NA	NA	NA
CaSee	12,878,114 KB	13,429,636 KB	21,388,220 KB	118,324,804 KB	120,300,268 KB
CopyKAT	707,728 KB	3,871,888 KB	25,668,260 KB	NA	NA
ikarus	298,576 KB	1,113,896 KB	12,877,444 KB	165,325,312 KB	165,368,696 KB

'NA' indicates that the method could not run correctly on the data.

6. In Figure 5c, the predicted cancerous spots by Cancer-Finder exhibit a more dispersed distribution compared to the pathological annotations. Will Cancer-Finder perform better if the authors incorporate spatial location information into the model.

Authors' response: We thank the reviewer's valuable advice.

The original **Figure 5c** has been removed in the revised version referring to the reviewer #3. However, in order to explore the impact of spatial information on prediction, we attempted to replace the linear layers in Cancer-Finder with graph convolutional layers to provide spatial information. We utilized the Voronoi diagram to construct the Graph based on the spatial information, where each node represents an ST spot and each edge represents the spatial neighborhood information between two

ST spots. All other network structures and training data are identical to the initial network. Following the same training with Cancer-Finder, we obtained a classifier and assessed it using four slides (1 BRCA, 1 CRC, 1 HCC, and 1 RCC). As shown in **response Figure 1**, the overall performance of the model is comparable to the original prediction outcomes. However, when spatial location information is considered, the model's prediction results become more spatially continuous but also introduce some false negatives.

We are grateful to the reviewer for this insightful suggestion, and we hope to investigate this model in greater depth in our future work.

Response Figure 1. Performance comparison of Cancer-Finder with and without spatial location information incorporated into the model. Below each figure, the accuracy of each result, as determined by pathologist-annotated outcomes, was displayed.

7. The methodological description of Cancer-Finder, particularly the introduction of the model in Figure 1, requires further clarification. Important concepts like "domains" need to be clearly defined, and the algorithmic foundation leading to Cancer-Finder's higher accuracy and speed compared to other tools should be elucidated.

Authors' response: We thank the reviewer's valuable advice. Typically, a "domain" refers to the specific data type or category on which a model is trained. Here, we apply this concept to the annotation of cellular malignant states in single-cell or spatial data by assuming that data from different tissues arise from different domains. We've added this clarification to the Results section (line 116-121, 126-136, page 5) and the Figure Legend of revised **Figure 1** to make the article more understandable. We also added some methodological description in the **Methods** (lines 481-507, page 15-16).

We believe that Cancer-Finder is more accurate because deep learning models are more adaptable and have a greater capacity for fitting than traditional models such as logistic regression (*Neural Netw.*, 1989, 2(5):359-366 and *J. Biomed. Inform.*, 2002,

35(5-6):352-359). In addition, Cancer-Finder makes effective use of accumulated cancer tissue data and annotation information (primarily through algorithmic calculations and manual annotations), thereby increasing the chance of accurately distinguishing between malignant and nonmalignant cells. While the majority of existing algorithms are based on simple models or single-dataset analyses, the former is susceptible to model limitations, whereas the latter is susceptible to the quality of the focused dataset and the cell type it contains.

We believe that Cancer-Finder is superior in terms of computational speed because it carries out prediction based on the classifier obtained from pre-training, and the time is linearly correlated with the number of cells. Its inference process requires only a simple forward-propagating linear computation, unlike other methods (CopyKAT must infer CNV and classify based on CNV profiles, SCECAN need to characterize the clonal structure and CaSee must find a reference to train). It is worth noting that the logistic regression-based method, *ikarus*, also has a theoretically fast inference speed, but its published pre-trained models lack accuracy. Retraining it is time-consuming and may also lead to suboptimal performance on certain datasets.

These discussions have been added to the revised **Discussion** section (lines 406-418 page 13).

Revised Figure 1. Overview of Cancer-Finder and its application. Cancer-Finder is a scalable framework that uses single-cell sequencing data to accurately annotate the malignant status of cells and is easily extensible to annotate other data (e.g. ST). The pre-trained model accurately and quickly identifies malignant cells derived from cancerous tissues. To counteract differences among different tissues, Cancer-Finder employs a domain generalization training strategy to improve general discriminatory

performance and accurately identify malignant cells in unexplored domains. Typically, a "domain" refers to the specific data type or category on which a model is trained. Here, we apply this concept to the annotation of cellular malignant states in single-cell or spatial data by assuming that data from different tissues arise from different domains. The model is a neural network composed of an input layer and two hidden layers for feature extraction and a layer for classification. Cancer-Finder uses risk extrapolation for domain generalization. This optimizes the model for high accuracy in all tissues because reducing risk differences across training domains can decrease a model's sensitivity to a broad range of extreme distribution shifts. In order to evaluate the performance of the model across multiple domains, this method minimizes two types of global risks: variance risk and average risk.

8. The authors should demonstrate the potential application of Cancer-Finder more effectively. The current application of Cancer-Finder to clear cell renal cell carcinoma (ccRCC) in Figure 5 seems unrelated to showcasing its performance. Instead, applying Cancer-Finder to larger datasets to highlight its computational speed and accuracy would be more relevant.

Authors' response: We thank the reviewer's insightful comment. We have added a new application that uses Cancer-Finder to annotate scRNA-seq data from cancerSCEM (*Nucleic Acids Res.*, 2022, 50(D1): D1147–D1155). Over 500,000 cells can be quickly predicted by Cancer-Finder in one hour. In addition, we confirmed the high accuracy of Cancer-Finder.

Detailly, since cancerSCEM only provides the percentage of malignant cells on each dataset and does not provide annotation information for each individual cell, we compared the percentage of malignant cells predicted by Cancer-Finder with the percentage provided by the database. The two are highly correlated (Pearson's correlation coefficient > 0.85 , Two-tailed p-value < 0.0001). This result demonstrates that Cancer-Finder can obtain highly consistent results with complex annotation strategies (combining manual and algorithmic annotations), proving its accuracy. This result has been added to the revised manuscript (lines 271-281 page 9)

On the basis of the annotation results, we also performed differential gene expression analyses between malignant and non-malignant cells of each cancer to identify candidate marker genes. The results are presented in revised **Figure 3e**, where it can be seen that the marker candidates identified using this strategy are extremely consistent with available knowledge. For instance, *EPCAM*, *KRT8*, *KRT18* are widely employed markers for malignant cell or cancer stem cell across various malignancies such as cervical cancer (*J. Med. Virol.*, 2023, 95(6):e28857), breast cancer (*Cell Death Discov.*, 2021, 7(1):104) and head & neck cancer (*Nat. Commun.*, 2023, 14(1):1055), and *SOX2* was considered as a cancer markers in glioblastomas (*Protein Cell*, 2023, 14(2):105-122) and esophageal squamous cell carcinoma (*EBioMedicine*, 2021, 69:103459). This result has been added to the revised manuscript (lines 282-293 page 10)

Notably, we have also previously used Cancer-Finder after training for prediction of TISCH data and found that Cancer-Finder performed poorly on some datasets. Thus, we compared the database annotation results to the original studies and found significant discrepancies. For example, for the tumor with lowest accuracy (STAD), we compared the database annotation results to the original study (*Cell Rep*, 2019, 27(6), 1934–1947) and found significant discrepancies. Detailly, nine patients with Non-Atrophic Gastritis (NAG), Chronic Atrophic Gastritis (CAG), Intestinal Metaplasia (IM), or Early Gastric Cancer (EGC) provided samples for the STAD dataset (GSE134520). According to the annotations of the TISCH database, malignant cells are present in the samples of patients with NAG and CAG, and IM diseases, but not in the samples of patients with early gastric cancer (EGC). However, according to the original study, patients with NAG and CAG should not have malignant cells in their samples, whereas patients with EGC should have many malignant cells in their samples. This discrepancy is likely the result of errors introduced by the database during the data collection process. This merely demonstrates that our tool can be a useful aid for relevant database annotation and error detection. All of these results and discussions have been added to the revised **Supplementary Figure 6**.

Revised Figure 3d-e. d, Comparison of Cancer-Finder's memory consumption to the other four methods. 'NA' indicates that the method could not run correctly on the data. e, Up-regulated genes identified from predicted malignant cells. Full names of the cancers are detailed in **Supplementary Table 15**.

Revised Supplementary Figure 6. Results of leave-one-cancer-type-out and leave-datasets-out cross-validations. a,b Results of leave-one-cancer-type-out cross-validation on single-cell data and balanced single-cell data. Leave-one-cancer-type-out cross-validation was performed by excluding scRNA-seq data from one cancer, training Cancer-Finder with data from other cancers, and predicting cell annotation labels for the excluded cancer. Cancer-Finder performs well (accuracy > 0.8) on most cancers, but its performance is limited on hematologic tumors (colored in grey), possibly due to the significant difference between hematologic and solid tumors. Notably, a discrepancy was discovered between the original study (*Cell Rep*, 2019, 27(6), 1934–1947) and TISCH's annotation on cancer with the lowest accuracy (colored red), which may have been caused by database collection errors in the database. c,d Results of leave-datasets-out cross-validation on single-cell data and balanced single-cell data. Leave-datasets-out cross-validation was conducted by excluding one scRNA-seq dataset at a time, training Cancer-Finder with data from other datasets, and predicting cell annotation labels for the excluded dataset. On four datasets (colored red), discrepancies were discovered between the original study (*Science*, 2020, 367(6476):405-411, *Cell Rep.*, 2019, 27(6), 1934–1947, *Cancer Discov.*, 2019,9(12):1708-1719, *Nat. Commun.*, 2020, 11(1):2285) and TISCH's annotation, possibly due to database collection errors. These results demonstrated that Cancer-Finder can be a useful aid for relevant database annotation and error detection. The datasets represented by the green bars consist of either all-malignant or all-non-malignant cells and are therefore not included in the balanced single-cell data.

9. Some tumor samples exhibit obvious pre-cancerous states in addition to normal and cancerous regions, which attract significant attention from the researchers in the field. Can the author expand the application scope of Cancer-Finder to identify pre-cancerous states in spatial transcriptomics?

Authors' response: We thank the reviewer's insightful comment. Unfortunately, pre-cancerous state is not annotated in Cancer-Finder's training data, making it difficult for Cancer-Finder to predict this state. This type of annotation is also extremely limited at present. We also attempted to predict it directly using the results of our current training, but the results were not ideal.

10. To generate broader interest in their application, the authors are encouraged to utilize more publicly available data to establish pre-trained models for pan-cancer single-cell transcriptomics and spatial transcriptomics.

Authors' response: We thank the reviewer's valuable advice. We gathered the data again, retrained the model, and uploaded them to our GitHub repository in order to expand the range of potential applications. In this revision, we collected 14 ST slides from 6 cancers, retrained the model, and uploaded the trained model to GitHub (Notably, currently, there is a dearth of accurately annotated available ST datasets). In the meantime, we retrained a classifier for single cell data using all the labeled single cell data collected in the article (476,562 cells from 13 different tissues) and uploaded to GitHub. Moreover, we will continue to collect new data to train more robust models in the future, which will be uploaded to github (<https://github.com/Patchouli-M/SequencingCancerFinder>).

11. To improve the usability of the tool, the authors should enhance the readability of the tutorial and code. Specific suggestions include providing more detailed tutorials, describing the input data format and output, offering additional options for input data formats (e.g., anndata format), and carefully reviewing the source code of Cancer-Finder to identify and rectify any errors.

Authors' response: We thank the reviewer's valuable advice. In this version, the tutorial and code have been carefully revised and expanded, and the Cancer-Finder source code has been thoroughly reviewed.

12. There are some errors in the code file, such as the help description of parameters "label_str" being the same as "gene_num" in the "get_args" function of the "args_utils.py" file. The authors should review the code thoroughly to correct such discrepancies.

Authors' response: We appreciate the careful review of the reviewer and apologize for the errors. In this version, the tutorial and code have been carefully reviewed and corrected.

Minor concerns:

1. The color differences corresponding to the tissues in figure 2a are too small, making it difficult to distinguish the tissue types in figure 2b and figure 2g.

Authors' response: We thank the reviewer's valuable advice. We adjusted the color of Figure 2, and the modified Figure is as follow:

Revised Figure 2. Performance evaluation of Cancer-Finder. a, Training data structure. In this study, 328230 cells from 13 distinct tissues were finally utilized. b, Visualization of internal validation data from 13 tissues utilizing t-SNE based on highly variable genes, wherein each point on the graph represents a single cell, each color represents a tissue, and each tissue has the same color as in (a). c, Similar to (b), the t-SNE visualization of data from 13 tissues by highly variable genes, colored according to malignant status. Red represents malignant cells and gray represents normal cells. d, T-SNE visualization utilizing highly-variable, neural network-transformed features and colored according to malignant status. Red represents malignant cells and gray represents normal cells. e, Changes in average risk throughout the training process. The training was carried out five times. f, Changes in variance risk throughout the training process. The training was carried out five times. g, Changes in the accuracy of prediction for different datasets from different tissues (13 tissues, using the same colors

as in (a)) throughout the training process. h, Accuracy of the pre-trained models. Accuracy of each tissue, including in-ternal validation data for 13 tissues and external validation data (mixed cell lines and PBMC, colored in purple). 5-fold leave-cells-out cross-validation was applied.

2. Please comment on the worse performance of bone dataset in Figure 2h.

Authors' response: We thank the reviewer's valuable advice. Multiple myeloma (MM) is a type of hematologic cancer. The dataset used in this study, referred to as "Bone" is comprised of samples obtained from patients with MM. Cross-validation, as displayed in revised **Supplementary Figure 6**, revealed that the performance of Cancer-Finder when applied to hematologic tumors was significantly suboptimal. We attempted to train a model using only data on hematologic tumors, but the results were unsatisfactory (revised **Supplementary Figure 14**). Therefore, we have included a recommendation regarding the inadequacy of the Cancer-Finder for hematologic tumors in our revised version (lines 437-441 page 13).

Revised Supplementary Figure 6. Results of leave-one-cancer-type-out and leave-datasets-out cross-validations. a,b Results of leave-one-cancer-type-out cross-validation on single-cell data and balanced single-cell data. Leave-one-cancer-type-out cross-validation was performed by excluding scRNA-seq data from one cancer, training Cancer-Finder with data from other cancers, and predicting cell annotation labels for the excluded cancer. Cancer-Finder performs well (accuracy > 0.8) on most cancers, but its performance is limited on hematologic tumors (colored in grey), possibly due to the significant difference between hematologic and solid tumors. Notably, a

discrepancy was discovered between the original study (*Cell Rep*, 2019, 27(6), 1934–1947) and TISCH's annotation on cancer with the lowest accuracy (colored red), which may have been caused by database collection errors in the database. c,d Results of leave-datasets-out cross-validation on single-cell data and balanced single-cell data. Leave-datasets-out cross-validation was conducted by excluding one scRNA-seq dataset at a time, training Cancer-Finder with data from other datasets, and predicting cell annotation labels for the excluded dataset. On four datasets (colored red), discrepancies were discovered between the original study (*Science*, 2020, 367(6476):405-411, *Cell Rep.*, 2019, 27(6), 1934–1947, *Cancer Discov.*, 2019,9(12):1708-1719, *Nat. Commun.*, 2020, 11(1):2285) and TISCH's annotation, possibly due to database collection errors. These results demonstrated that Cancer-Finder can be a useful aid for relevant database annotation and error detection. The datasets represented by the green bars consist of either all-malignant or all-non-malignant cells and are therefore not included in the balanced single-cell data.

Revised Supplementary Figure 14. Performance of Cancer-Finder trained specifically on hematologic tumors. Here, we evaluate the model's accuracy on four hematologic cancers using the leave-one-cancer-type-out strategy.

Reviewer #3 (Remarks to the Author): Expert in cancer genomics, renal cancers, single-cell RNA-seq, spatial transcriptomics, and tumour microenvironment

Zhong et al present Cancer-Finder: Domain generalization enables general cancer cell annotation in single-cell and spatial transcriptomics. In general, the tool could be of use for the cancer genomic community, and they have assessed the tool with a large number of single cell and spatial datasets. The ability of a tool to discern cancer cells, and those of the stroma and immune system from single cell and ST data, would be useful.

However, I have concerns about the evidence provided. This principally centres around nomenclature and the ability of the tool to determine cell state when there are admixed normal cells. The ST data presented, especially that for ccRCC data, does not have any non-tumour regions, and for these to be annotated as such is misleading. There will be immune infiltrates that could be immune niches, but importantly they are part of the overall tumour structure. The authors need to test the model on systems that either have infiltrative cancer types, or slides that traverse the tumour normal interface. The authors might have already done this, but due to the poor annotation of the slides provided in the manuscript, it is impossible to assess.

Authors' response: We are grateful for the reviewer's positive comments on the novelty and significance of our study, as well as for pointing out the errors in the algorithm application part. As suggested by the reviewer, we downloaded the ST slides from [https://www.cell.com/cancer-cell/fulltext/S1535-6108\(22\)00548-7](https://www.cell.com/cancer-cell/fulltext/S1535-6108(22)00548-7), made predictions, and redid the biological exploration.

Detailly, using 5 ccRCC 10x Visium slides containing tumor-normal interfaces, Cancer-Finder was used to predict spot status (malignant or non-malignant), determine malignant and histologically normal tissues on ST slides based on the prediction results, investigate features that differentiate malignant spots with non-malignant spot, and investigate the relationship between these features and prognosis in order to provide insight into the microenvironment of ccRCC.

The results revealed that Cancer-Finder has superior or comparable capabilities in identifying malignant spots and does not require a priori information and data compared to conventional markers or deconvolution techniques. Simultaneously, the model interpretability module (a new module of Cancer-Finder added in this version based on the recommendation of reviewers #4 and #5) revealed a gene signature consisting of ten genes (the majority of them were reportedly active participants in the EMT process in a variety of cancers.) that tends to be enriched at the tumor-normal interface, may be associated with the formation of an invasive tumor microenvironment, and is a better prognostic indicator than the available EMT programs (the Hallmark EMT pathway and the ccRCC EMT program). These findings deserve further study on the function of the enriched regions of this signature (especially at the tumor-normal interface) and its association with prognosis, which may lead to a better understanding of the renal cancer

microenvironment. This new application demonstrated the considerable potential of Cancer-Finder for application in studies associated with tumor microenvironments.

Sincerely appreciate the reviewer's extremely professional comment regarding our application section, which offered valuable guidelines for enhancing our article.

Lines 42-43: The poor correlation is disputed, particularly in the light of IO agents, and regardless is not supported by data in this study. Having this statement at the end of the abstract is misleading as it infers that this is a conclusion you have drawn from your data.

Authors' response: We thank the reviewer's valuable advice. We have removed this part of the results from this paper, as well as this conclusion, from the abstract.

Lines 44-45: I am not convinced that you have discovered any new biological insights. Please justify or remove.

Authors' response: We thank the reviewer's valuable advice. We have removed this part.

Lines 59-60: You have missed the ST deconvolution methods that use a reference to determine the cellular makeup of tissue such as cell2location.

Authors' response: We thank the reviewer's valuable advice. We have added more discussion on ST deconvolution methods at line 87, page 4 (in the current version). Due to the fact that Cancer-Finder can be used for both single-cell and ST data, lines 59-60 (where it was in the previous version) are focused primarily on the introduction of the methods applicable to single-cell data, while the methods applicable to ST are not mentioned until the fourth paragraph of the Introduction.

Lines 97-99: The ST slides you have tested on are all cancerous, ie there is an admixture of cancer cells, immune cells, and stromal cells. Measuring the effectiveness of the tool between these types is a rather low barrier. I would suggest that a better comparison would be to test the model using samples that include the normal cells that make up the cell of origin for the cancer cells. This is because the cell of origin may be the cells most transcriptionally similar to the cancer. The test set could be samples where the cancerous tissue is admixed with normal epithelial cells, such as prostate cancer <https://www.nature.com/articles/s41586-022-05023-2>, or samples where the tumour normal interface was included [https://www.cell.com/cancer-cell/fulltext/S1535-6108\(22\)00548-7](https://www.cell.com/cancer-cell/fulltext/S1535-6108(22)00548-7). This is a particular concern as your training set does not contain either benign or malignant cells of these tissue types. Without this comparison, the performance of your method cannot be assessed.

Authors' response: We thank the reviewer's valuable advice. We appreciate the reviewer for including links to the appropriate data. We used 5 ST slides that we downloaded from [https://www.cell.com/cancer-cell/fulltext/S1535-6108\(22\)00548-7](https://www.cell.com/cancer-cell/fulltext/S1535-6108(22)00548-7) to make predictions. Cancer-finder works well on samples containing tumour normal

interface, as shown in **Revised Figure 5**. This section of the results has been added to the revised Results (lines 333-388 page 10-12).

Revised Figure 5. Application of Cancer-Finder on intertumor heterogeneity analysis in ccRCC ST dataset. a, Annotations of malignant regions obtained by different methods, including prediction by Cancer-Finder (first left), annotation by pathologists (second left), deconvolution by CARD (middle), gene expression of CA9 (a marker of ccRCC, second right) and gene expression of ANGPTL4 (a marker of ccRCC, first right). b, Co-localization of genes in the gene signature (blue) and prediction of malignant area (red). The co-localization state was assigned to the spot when nine of the ten genes in the signature have expression values (UMI count > 0). The result of malignancy prediction is represented by the softmax value, with the

likelihood of being malignant increasing as the value rises. Where 0.50 is the default softmax cutoff value. c, The proportions of the top ten representative cell types in spots with a co-localization state across 5 slides. d, Gene expression of genes in the detected gene signature across cells. e, Survival analyses using the ssGESA score of the genes of Hallmark EMT pathway. f, Survival analyses using the ssGESA score of the gene signature from Cancer-Finder. g, Survival analyses using the ssGESA score of the genes of ccRCC EMT program. Log-rank test was applied.

Lines 159-163: As before, separating stromal and immune cells from malignant cells is perhaps the cell classification. However, this plot does not address the question of classifying cancer cells from their normal cell of origin.

Authors' response: We appreciate the reviewer's comment. Because the TISCH database contains only annotations for stromal, immune, and cancer cells, normal epithelial cells cannot be displayed in this figure. To characterize this, we downloaded single-cell sequencing data from non-cancerous patients from The Tabula Sapiens database (*Science*, 2022, 376(6594):eab14896.) and added them to the training set, and plotted t-SNE on the validation data. To distinguish malignant cells from normal epithelial cells more effectively, we trained a Cancer-Finder classifier that is tri-classified (other cells, cancer cells, and normal epithelial cells, respectively). As shown in response Figure 3, after the feature transformation of the Cancer-Finder network, the corresponding t-SNE visualization reveals that these three types of cells can be completely separated, whereas before the transformation, they cannot be completely separated.

A recently published database (3CA, *Nature*, 2023, 618(7965):598-606) contains more extensive annotations regarding normal epithelial cells than TISCH. In the future, we will endeavor to gather more comprehensive databases and utilize the data derived from these sources to enhance the training of a classifier. Subsequently, we intend to share the outcomes on GitHub.

Response Figure 3. Cell distribution according to t-SNE visualizations. Cell distribution according to t-SNE visualization based on 2000 HVGs and colored in tissue labels (a) and cell types (b). c, Cell distribution according to t-SNE visualization based on transformed features from Cancer-Finder and colored in cell types.

Lines 261-262: The plots are poorly labelled and the colours not annotated. It should be clear what the pathologists have labelled, especially in the 'normal' areas - do these have normal architecture or are they simply regions of tumour that have high stromal or immune infiltration?

Authors' response: We apologize for the confusion here. We have modified **Figure 4c-d.** and added color annotations. These H&E images with their malignant area annotation were derived from existing studies (BRCA: <https://www.10xgenomics.com/resources/datasets/human-breast-cancer-block-a-section-1-1-standard-1-1-0>, OV: <https://www.10xgenomics.com/resources/datasets/human-ovarian-cancer-whole-transcriptome-analysis-stains-dapi-anti-pan-ck-anti-cd-45-1-standard-1-2-0>, HCC and ICC: <https://ngdc.cncb.ac.cn/gsa-human/browse/HRA000437>; CRC: <https://ngdc.cncb.ac.cn/gsa-human/browse/HRA000979>; ccRCC: <https://www.ncbi.nlm.nih.gov/geo/query/acc.cgi?acc=GSE175540>). The annotations for the non-malignant and malignant area were referenced from a previous study (*Nat. Commun.*, 2023, 14(1):922) and revised by pathologists, and some slides without available annotation were manually annotated by pathologists directly. Due to the limited resolution of these publicly available images, however, pathologists were only able to separate malignant tissue areas but were unable to annotate the finer structures. We sincerely apologize, but the available data do not permit us to annotate in greater detail at this time.

Revised Figure 4. Expansion of Cancer-Finder to spatial transcriptomic annotation. a, Expansion of the training process to include ST data. Annotations from pathologists were used to determine which spots were malignant. Training is completed by replacing the single-cell matrix with the spot expression matrix directly. b, Specifics regarding the datasets used during spatial transcriptome training and testing, as well as the accuracy rates obtained. c, Comparison of model predictions and annotation labels

(left) from pathologists for cases where the tested cancer type was included (middle) or excluded (right) from the training data. d, Comparison between model predictions and expression of malignant marker genes on MERFISH data (up), and comparison between model predictions and annotations from original research on Slide-seq data (down).

Lines 282-283: This association is not clear, and there is conflicting evidence in the literature.

Authors' response: We appreciate the reviewer's comment. We have removed this part in this revision.

Line 290: This is misleading. The entire biopsy shown is tumour, but some areas have higher immune infiltration. It is all tumour region!

Authors' response: We apologize for the misleading description and appreciate the reviewer's comment. In the new version, the names of these regions have been carefully corrected. We used 10x Visium slides containing tumor and histologically normal tissues adjacent to the tumors from [https://www.cell.com/cancer-cell/fulltext/S1535-6108\(22\)00548-7](https://www.cell.com/cancer-cell/fulltext/S1535-6108(22)00548-7). Spots are labeled as malignant spots and non-malignant spots.

Lines 311: again, these are not "non-tumour" regions.

Authors' response: We apologize for the misleading description and appreciate the reviewer's comment. In the new version, the names of these regions have been carefully corrected.

Reviewer #4 (Remarks to the Author): Expert in single-cell RNA-seq and spatial transcriptomics methods and analysis, bioinformatics, and deep learning; co-reviewed with Reviewer #5

Summary: Zhong and colleagues introduce "Cancer-Finder," a domain generalization-based deep-learning algorithm that identifies cancer cells from single-cell data with an average accuracy of 91.60%. The authors claim that cancer-finder achieves high accuracy and generalizes well across different tissues and cancer types. The authors then attempt to apply Cancer-Finder on 23 ccRCC ST and observe evidence that shows gene expression patterns related to inflammation and CD8+ T cell exhaustion in cancerous spots, providing valuable insights for cancer research and diagnosis.

The paper's topic is crucial, showing promising results, but lacks sufficient evidence for Cancer-Finder's generalization. Rigorous cross-validation using leave-datasets-out methodology, detailed data preprocessing explanations, and interpretability efforts are necessary for credibility. Comprehensive evaluation with various metrics, including sensitivity analysis, is crucial. Without addressing these issues, the study's claim of a generalized model is underpowered, hindering its impact on the cancer annotation community.

Authors' response: We are grateful for the reviewer's positive comments on the novelty and significance of our study. Additionally, we appreciate the reviewer's significant comments on how to improve the model validation in our work. In this version, we've included additional rigorous cross-validation, detailed data preprocessing explanations, a comprehensive evaluation with multiple metrics, interpretability efforts, and sensitivity analysis.

Major Critiques:

1- Spatial Transcriptome Data Acquisition Technology: Spatial transcriptome data acquisition platforms, like Spatial Transcriptomics, Slide-seq, Visium, MERFISH, and STARmap, have specific strengths and limitations impacting data quality, resolution, and sensitivity. Rigorous evaluation with multiple datasets and cross-validation is crucial to assess model robustness and identify potential biases from platform variability. The paper should highlight the used platform, address biases, and demonstrate the impact of domain adaptation and transfer learning on generalization. Developing the model on only one platform risks lacking "generalizability".

Authors' response: We appreciate the reviewer's comment.

Other than the commercial platform 10X Visium, which has been utilized in a variety of applications, other platforms have fewer use cases, and even fewer data on cancer tissues. In this situation, it is challenging to collect a large enough training set (including at least 2-3 types of cancer data) to train a pre-trained model on data from multiple platforms. Consequently, this revision focuses primarily on predicting data

from other platforms using the training results from the existing training set (the pre-trained models based on scRNA-seq data and 10X Visium data). Here, we primarily focus on making predictions using datasets from one imaging-based technique (MERFISH, *Science*, 2015, 348(6233):aaa6090) and two sequencing-based techniques with different resolutions, namely Slide-seq (*Science*, 2019, 363(6434):1463-1467) and legacy ST (*Science*, 2016, 353(6294):78-82). Detailly, four MERFISH slides (<https://info.vizgen.com/ffpe-showcase>), four Slide-seq slides (*Cell*, 2022, 185(14):2591-2608) and two legacy ST slides (*Cell*, 2020, 182(6):1661-1662) were downloaded.

Considering that MERFISH data are most similar to the single-cell form, we initially trained the model with scRNA-seq data. Here, we trained the model using the single-cell sub-matrix (containing 550 genes measured in the MERFISH data), and utilized the training results to predict malignant cells in the MERFISH dataset. As shown in the revised **Supplementary Figure 7a** and **Figure 4**, in the case of using a suitable Softmax threshold, Cancer-Finder has a high degree of accuracy on the MERFISH data. Notably, we observed that Cancer-Finder may generate false positives when the pre-trained model was applied directly to MERFISH data with the default softmax threshold (threshold = 0.5) because single-cell data and MERFISH data are not identical. Based on a MERFISH slide, the ROC curve was used to determine the optimal threshold (threshold = 0.9766), Cancer-Finder was able to accurately predict MERFISH data (accuracy: 70.69–83.84 %, AUC: 0.7707–0.8969).

Similarly, we have expanded our predictions to Slide-seq data. This is a second-generation sequencing-based ST technology with near single-cell resolution (spot diameter of 10um), so we still made predictions with the pre-trained model we obtained on the scRNA-seq dataset, and the results demonstrated that Cancer-Finder performs exceptionally well on the majority of the datasets (Revised **Supplementary Figure 7b**).

Lastly, we attempted to extend the model to legacy ST slides with a larger spot (spot diameter of 100um) and made predictions utilizing a pre-trained model trained on 10x Visium slides. As shown in revised **Supplementary Figure 7c**, the performance of Cancer-Finder varies across datasets (slide 1: accuracy=0.8050, AUC=0.8227; slide 2: accuracy=0.5765, AUC=0.5650).

In conclusion, it is difficult for us to provide a good pre-trained model for every type of ST data given the current state of data accumulation. Notwithstanding, the prediction results of models trained on existing scRNA-seq and 10x Visium ST slides on sequencing data from platforms with comparable resolution are reasonably accurate. In the future, we will attempt to collect and accumulate additional data for such an exercise. We have added these discussions to the revised Result (lines 319-332 page 11), Discussion (lines 424-427 page 14), and SI (revised **Supplementary Note 5 and Supplementary Figure 7**) parts.

The generalization capability of Cancer-Finder was also evaluated on single-cell data by removing data from a specific technique from the training set and making predictions on this data. Cancer-Finder is able to generalize effectively across techniques (with an average accuracy of 86.35%), but its accuracy fluctuates among

different techniques. We have incorporated this discussion into the revised **Discussion** (lines 395-398, page 13), and SI (revised **Supplementary Figure 10**) sections, aiming to assist users in making informed decisions.

Revised Figure 4. Expansion of Cancer-Finder to spatial transcriptomic annotation. a, Expansion of the training process to include ST data. Annotations from

pathologists were used to determine which spots were malignant. Training is completed by replacing the single-cell matrix with the spot expression matrix directly. b, Specifics regarding the datasets used during spatial transcriptome training and testing, as well as the accuracy rates obtained. c, Comparison of model predictions and annotation labels (left) from pathologists for cases where the tested cancer type was included (middle) or excluded (right) from the training data. d, Comparison between model predictions and expression of malignant marker genes on MERFISH data (up), and comparison between model predictions and annotations from original research on Slide-seq data (down).

Revised Supplementary Figure 7. Application expansion of Cancer-Finder. a, Performance of Cancer-Finder on MERFISH data. Here, the optimal Softmax threshold (threshold = 0.9766) was determined according to the ROC curve based on an external MERFISH slide. b, Performance of Cancer-Finder on slide-seq data. c, Performance of Cancer-Finder on “ST” data. *’ denotes the slide used to determine the threshold value.

Revised Supplementary Figure 10. Results of leave-platform-out cross-validations. Leave-platform-out cross-validation was performed by excluding datasets from one platform, training Cancer-Finder with data from other platforms, and predicting cell annotation labels for the excluded datasets. Each point represents the accuracy of a dataset. The dataset from mCEL-seq2 was excluded from the balanced validation because it contains only non-malignant (negative) cells. Notably, several datasets were excluded from balanced validation because they contained only all-malignant or all-nonmalignant cells. The boxes are centered at median values, where the range of boxes represents the interquartile range (IQR) bounded by the first quartile (Q1) and the third quartile (Q3).

2- Cross-validation Techniques: The paper introduces important findings, but there are concerns regarding the cross-validation techniques and the “generalizability” of the model. While the authors mention using five-fold cross-validation, there is a lack of detailed information on the process. It is essential to clarify whether they used leave-cells-out or leave-datasets-out cross-validation, as the former can lead to overestimating generalizability by "memorizing" unique signatures of individual datasets. To strengthen the paper's claims, I suggest repeating the analysis with leave-datasets-out cross-validation, as it can better assess domain generalization capabilities and highlight the model's performance across different cancer types. Addressing these issues would provide a more comprehensive understanding of the model's true generalizability and improve the study's impact in the field of cancer annotation. To rigorously assess the model's performance and domain generalization capabilities, the authors must conduct experiments using leave-datasets-out cross-validation. Additionally, investigating leave-one-cancer-type-out cross-validation would be essential, as it directly relates to the paper's claims of generalizability. Without addressing these issues, the study's credibility is at risk, and its claims about model generalization in the cancer annotation community lack solid evidence.

Authors' response: We appreciate the reviewer's comment. We apologize for that we did not perform these validations in the previous version. In the initial version, we solely conducted leave-cells-out cross-validation (revised **Figure 2h**). Leave-one-cancer-type-out (revised **Supplementary Figure 6 a-b**), leave-datasets-out (revised **Supplementary Figure 6 c-d**), and 5-fold leave-cells-out (revised **Figure 2h**) cross-validations are all included in this revision.

We have added these results to the revised **Results** (lines 277-281 page 9), and SI (revised **Supplementary Figure 6**).

Revised Supplementary Figure 6. Results of leave-one-cancer-type-out and leave-datasets-out cross-validations. a,b Results of leave-one-cancer-type-out cross-validation on single-cell data and balanced single-cell data. Leave-one-cancer-type-out cross-validation was performed by excluding scRNA-seq data from one cancer, training Cancer-Finder with data from other cancers, and predicting cell annotation labels for the excluded cancer. Cancer-Finder performs well (accuracy > 0.8) on most cancers, but its performance is limited on hematologic tumors (colored in grey), possibly due to the significant difference between hematologic and solid tumors. Notably, a discrepancy was discovered between the original study (*Cell Rep*, 2019, 27(6), 1934–1947) and TISCH's annotation on cancer with the lowest accuracy (colored red), which may have been caused by database collection errors in the database. c,d Results of leave-datasets-out cross-validation on single-cell data and balanced single-cell data. Leave-datasets-out cross-validation was conducted by excluding one scRNA-seq dataset at a time, training Cancer-Finder with data from other datasets, and predicting cell annotation labels for the excluded dataset. On four datasets (colored red),

discrepancies were discovered between the original study (*Science*, 2020, 367(6476):405-411, *Cell Rep.*, 2019, 27(6), 1934–1947, *Cancer Discov.*, 2019,9(12):1708-1719, *Nat. Commun.*, 2020, 11(1):2285) and TISCH's annotation, possibly due to database collection errors. These results demonstrated that Cancer-Finder can be a useful aid for relevant database annotation and error detection. The datasets represented by the green bars consist of either all-malignant or all-non-malignant cells and are therefore not included in the balanced single-cell data.

Revised Figure 2h. Accuracy of the pre-trained models. Accuracy of each tissue, including internal validation data for 13 tissues and external validation data (mixed cell lines and PBMC, colored in purple). 5-fold leave-cells-out cross-validation was applied.

3- Data Preprocessing: The paper briefly mentions data preprocessing steps but lacks in-depth explanations and the rationale behind these choices. Elaborating on the data preprocessing methods and their potential impact on results is crucial for better understanding the model's performance. The paper must provide a comprehensive and detailed explanation of the data preprocessing methods used, along with the rationale behind each choice. Understanding the impact of preprocessing on results is crucial for interpreting the model's performance accurately.

Authors' response: We thank the reviewer's valuable advice. We apologize for this omission in our original version; in this new version, we provide a thorough description of the data preprocessing, including the following sections: Dataset collection and merging, balanced sampling, Feature selection, model parameter determination, and cross-validation. These were added to the SI (revised **Supplementary Note 2**).

4- Interpretability of the Model: The paper acknowledges that the deep learning model can be hard to interpret. While this is expected, it is needed that the authors attempt to shed some light on the model's inner workings using techniques like saliency maps. This effort can enhance the credibility of the results and aid in clinical applications. Despite acknowledging the model's black box nature, the authors should make an effort to shed light on the model's inner workings using techniques like saliency maps. This will enhance the credibility of the results and enable potential clinical applications.

Authors' response: We are extremely grateful for the reviewer's insightful suggestions, which led to an important improvement of our model. Saliency Map (*arXiv*, 2013, arXiv:1312.6034) was utilized to determine the model's interpretability. Specifically, the gradient of the loss function can be obtained as fellow based on backpropagation:

$$W = \{w_1, w_2, \dots, w_m\}^T = \text{Gradient}(\text{Loss}(\theta))$$

where $\text{Loss}(\theta)$ is the loss function, W is the gradient containing m elements, and m is the number of features. $m = 5000$ for ST and $m = 4572$ for single cell data. After one training loop, the salience value for each feature (gene) was defined as follows:

$$\text{salience}_{\text{Gene } i} = \text{sum}(w_{\text{Gene } i}) = \sum_{e=1}^n |w_i^e|$$

where n is the number of samples within a training loop. The salience value indicates the contribution of each gene to the training. The genes were subsequently ranked based on their contributions during training. After the top ten genes remained unchanged for 20 epochs, it was concluded that the ranking of the genes, in terms of their contribution to the training, had reached a stable state. In our experiments, this occurred between the 69-89th training rounds, at which point we selected the ten most significant genes for co-localization and subsequent analysis. These were added to the SI (revised **Supplementary Note 1 and Supplementary Figure 8**).

Revised Supplementary Figure 8. Changes in the salience value of features during training. Here, the salience values for the top 20 genes are displayed.

5- Evaluation Metrics: F1 score and other metrics, such as ROC curves, are not discussed. It would be valuable to include a comprehensive evaluation of the model's performance using various metrics to provide a more comprehensive assessment. The paper should include a thorough evaluation of the model's performance using various metrics, including F1 score and ROC curves. This comprehensive assessment will provide a more robust understanding of the model's strengths and weaknesses.

Authors' response: We thank the reviewer's valuable advice. In this new revision, we have added precision, recall, F1 scores, AUROC and AUPRC values to each dataset to provide a more complete picture of Cancer-Finder's performance (revised **Supplementary Figure 2-5 and Figure 3b**).

During algorithm comparisons, we included accuracy, precision, recall, and F1 scores. We were only able to calculate the AUROC and AUPRC values for Cancer-Finder because other algorithms did not provide a continuous value (such as the Softmax value in Cancer-Finder) that could be used for these two metrics.

Revised Figure 3b. Comparison of Cancer-Finder's prediction accuracy to four other cell annotation algorithms. Cancer-Finder demonstrated significantly greater accuracy and similarity (to the labels from the original studies) than other tools across various cancers. All tests were conducted in parallel five times. The presence of an 'NA' denotes that the method returns an error and cannot be executed with these data.

Revised Supplementary Figure 2. Comparison of the Precision of Cancer-Finder and four other cell annotation algorithms. Each test was conducted five times in parallel. The presence of an ‘NA’ denotes that the method returns an error and cannot be executed with these data, or that the dataset contains only positive or negative samples, therefore the indicator cannot be calculated.

Revised Supplementary Figure 3. Comparison of the F1 score and the Recall rate of Cancer-Finder and four other cell annotation algorithms. Each test was

conducted five times in parallel. The presence of an 'NA' denotes that the method returns an error and cannot be executed with these data, or that the dataset contains only positive or negative samples, therefore the indicator cannot be calculated.

Revised Supplementary Figure 4. AUROC of Cancer-Finder's prediction results across test datasets. Each test was conducted five times in parallel, and the average AUROC was presented.

Revised Supplementary Figure 5. AUPRC of Cancer-Finder's prediction results across test datasets. Each test was conducted five times in parallel, and the average AUPRC was presented.

6- Bias and Sensitivity Analysis: The use of a 1-to-1 ratio in cancer cell classification is commendable to prevent bias towards specific cell types. However, the paper lacks sensitivity analysis, especially for spatial data. Conducting

simulations and providing insights into the model's sensitivity to different input dimensions would strengthen the paper's conclusions. The use of a 1-to-1 ratio for cancer cell classification is commendable, but the paper must conduct sensitivity analysis, especially for spatial data. Simulations should be performed to gain insights into the model's sensitivity to different input dimensions, ensuring the model's reliability and generalizability.

Authors' response: We thank the reviewer's valuable advice. We have performed a sensitivity analysis on the ratio of positive to negative samples, as shown in the revised **Supplementary Figure 16**. In particular, malignant or non-malignant cells (or spots) were sampled by down-sampling in each domain to produce a series of data with ratios ranging from 0.1:1 to 1:0.1. Then, 5-fold leave-cells-out cross-validations were performed to assess the performance of Cancer-Finder.

This result has been added to the revised **Methods** (lines 520-521, page 12) and revised **Supplementary Figure 16**.

Revised Supplementary Figure 16. Sensitivity analysis on the ratio of positive to negative samples. Malignant or non-malignant cells (or spots) were sampled by down-sampling in each domain to produce a series of data with ratios ranging from 0.1:1 to 1:0.1. Then, 5-fold leave-cells-out cross-validations were performed to assess the performance of Cancer-Finder (Error bars show mean \pm standard deviation of these 5 validations).

7- Ambiguous Feature Extraction Module: The description of the feature extraction module is ambiguous and lacks clarity. The number of neurons in each layer and the activation functions used in the fully connected layers should be clearly specified. Additionally, details about how the dropout probability was determined to prevent overfitting need to be provided.

Authors' response: We thank the reviewer's valuable advice.

As shown in Figure 1, Cancer-Finder's model consists of a feature extraction module and a classification module. Feature extraction module comprises two fully

connected layers, separated by a random dropout layer to prevent overfitting. Without the use of activation functions, the layers are connected directly. The first of fully connected layers employs the same number of neurons as the selected features after data processing (4572 for scRNA-seq data, 5000 for ST data), while the second fully connected layer reduces the number of neurons to 512, creating a bottleneck layer. The feature extraction module is connected to a classification module which consists of a classification layer to complete the classification task. The number of neurons in the classification layer corresponds to the number of classes required for the classification task, which in this case, is two.

The default dropout probability for the intermediate dropout layer was set to 0.5, as this value was recommended (*J. Mach. Learn. Res.*, 2014, 15(1), 1929–1958), and commonly employed in similar studies (*Nat. Med.*, 2019, 25(1):60-64, *Nat. Mach. Intell.*, 2022, 4(8):710-719).

We apologize for not making this clear in the previous version; this section has been expanded in this revised **Results** (lines 126-132 page 5) and revised **Methods** (lines 457-467 page 15).

8- Incomplete Training Approaches: The section on training approaches for domain generalization tasks lacks complete information. The equation provided for the loss function needs further elaboration, and the rationale behind using the proposed risk extrapolation method should be explained in detail. Additionally, the role of the hyperparameter β and how it affects the training process should be clarified.

Authors' response: We thank the reviewer's valuable advice. We have revised and expanded the description of the loss function to include an explanation of β 's function and setting:

The risk functions used in the training process of domain generalization were proposed by David Krueger et al (*ICML PMLR*, 2021:5815-5826) and termed risk extrapolation. This method serves two purposes: 1) to reduce training risk, and 2) to increase the similarity of training risk across domains to complete domain generalization. In this study, data from m different tissues in the training set are considered as different source domains and denoted as x_1, x_2, \dots, x_m . The data labels are denoted as y_1, y_2, \dots, y_m , and the feature extraction and classification network is denoted as f . Risks of m training source domains are represented as $\mathcal{R}_1, \dots, \mathcal{R}_m$. The cross-entropy loss function is used to represent the training risk of each domain x_e , for any $e \in \{1, 2, \dots, m\}$:

$$\mathcal{R}_e(\theta) = \text{CrossEntropy}(f_\theta(x_e), y_e)$$

To prevent excessively high values for the total training risk from multiple source domains, we use the mean of the domain risks \mathcal{R}_{Avg} to characterize the training risk:

$$\mathcal{R}_{Avg}(\theta) = \frac{1}{m} \sum_{e=1}^m \mathcal{R}_e(\theta)$$

To describe the discrepancy of training risk across domains, we use the variance of training risks across training domains as follows:

$$\mathcal{R}_{var}(\theta) = \frac{\sum_{e=1}^m (\mathcal{R}_e(\theta) - \mathcal{R}_{Avg}(\theta))^2}{m}$$

The loss function, which is proposed to reduce training risk while increasing the similarity (reducing the discrepancy) of training risk across domains, is described as follows:

$$L(\theta) = \beta * \mathcal{R}_{var}(\theta) + \mathcal{R}_{Avg}(\theta)$$

The first term of the equation controls the domain-to-domain risk variance, whereas the second term controls the total training risk. θ represents the model parameter and β is a hyperparameter to adjust the weights of the two risk terms. Detailly, β serves as an important hyperparameter in the risk extrapolation method, controlling the balance between reducing the average risk and enforcing equality of risks, with $\beta \rightarrow 0$ recovering ERM, and $\beta \rightarrow \infty$ leading to a complete concentration on making the risk equal, thereby completing domain generalization (*ICML PMLR*, 2021: 5815-5826). In this study, $\beta = 1$ was chosen to balance between the two risks (detailed in **Supplementary Note 2**: model parameter determination, and **Supplementary Figure 15**).

Detailly, using 5-fold cross-validation, several β values were accessed. β was set to 0, 0.2, 0.4, 0.6.....1.8, 2 and 4, 6, 8 for model training. Here, models were trained and evaluated using the scRNA-seq training set with 340,178 cells. Models training was terminated when the accuracy of breast cancer data prediction reached the maximum. As shown in **Supplementary Figure 15**, the most effective beta fluctuates around 1 (0.6-2) during 5-fold cross-validations. Considering that the larger the β , the less weight is given to the evaluation of the total training risk in the loss function, $\beta = 1$ was chosen to control the overall training risk (cross-entropy loss in this case).

Revised Supplementary Figure 15. 5-fold cross-validation of Cancer-Finder with various β .

In addition, we have added a discussion regarding the selection of risk extrapolation method to this task as fellow (revised **Supplementary Note 6**):

In cell classification and annotation, neural networks have numerous applications and perform exceptionally well (*Nat. Rev. Mol. Cell Biol.*, 2022, 23(5):303-304). Tumor heterogeneity creates genetical differences in the distribution of gene expression in different cancers (*Nat. Genet.*, 2020, 52(11):1208-1218), whereas neural networks are sensitive to distribution shift (*IEEE TEC*, 2019, 23(5):828-841). Domain generalization is specifically designed for this type of problem (*IEEE TKDE*, 2023, 35(8):8052-8072). Among domain generalization strategies, V-REx has a simple and efficient mathematical form, which makes its computation less complex and computationally burdensome, and therefore more suitable for training on large datasets. In addition, nine domain generalization strategies were evaluated by Wang et al. (*IEEE TKDE*, 2023, 35(8):8052-8072), and the evaluated results are available at github (<https://github.com/jindongwang/transferlearning/tree/master/code/DeepDG>) .

Based on the results, Variance Risk Extrapolation (V-REx) exhibits consistent and robust performance across four sets of evaluations on two datasets (PACS dataset and Home-Office dataset), consistently placing in the top three in three of these evaluations. Overall, we are confident that this approach can significantly enhance the annotation of the malignant state within the tumor microenvironment across various types of cancer. This discussion has been added in revised **Supplementary Note 6** and revised **Supplementary Table 17** in this revision.

[Redacted]

9- Data Collection and Processing: The description of data collection and processing is inadequate. The TISCH database is mentioned, but there is no information on the specific datasets used and their characteristics. Additionally, details on how the down-sampling of malignant and non-malignant cells was performed to achieve a 1:1 ratio need to be provided.

Authors' response: We thank the reviewer's valuable advice. As mentioned in response of comment#3, we have added a detailed description of the data preprocessing in **Supplementary Note 2** in SI. This part includes the following sections: Dataset Collection and Merging, Balanced Sampling, Feature selection, Model Parameter Determination, and Cross-Validation. Datasets from TISCH database are described in the "Dataset collection and merging" section. Details on down-sampling are included in the "Balanced Sampling" section.

10- Provided codes: The provided code seems relatively simple and concise. While simple and concise, the absence of comments or explanatory notes is a major flaw, hindering usability and maintainability. Thorough comments must be added to improve the code's quality and accessibility.

Authors' response: We thank the reviewer's valuable advice. We included detailed code descriptions and comments in this revision.

Minor Critiques:

Line 154: There is missing space after "." and "The" ((2a).The).

Authors' response: We apologize for this error and thank the reviewers for their careful reading.

Line 272: What type of "general predictions" do the authors mean?

Authors' response: We thank the reviewers for their careful reading. We wanted to show that Cancer-Finder can be used to predict accurately on other datasets after training on a small dataset. The phrase "general predictions" has been removed from the new version because it might not be appropriate.

Line 293: For Figures 5c and 5d, the figure's scale is unclear to me.

Authors' response: We thank the reviewers for their careful reading. The scale has been added to revised **Figure 5a-b** in this revised manuscript. It is important to note, however, that in response to reviewer 3's suggestion that the ST slide we have presented here may not completely validate our model, we have used new data to display a more trustworthy evaluation.

Revised Figure 5. Application of Cancer-Finder on intertumor heterogeneity analysis in ccRCC ST dataset. a, Annotations of malignant regions obtained by different methods, including prediction by Cancer-Finder (first left), annotation by pathologists (second left), deconvolution by CARD (middle), gene expression of CA9 (a marker of ccRCC, second right) and gene expression of ANGPTL4 (a marker of ccRCC, first right). b, Co-localization of genes in the gene signature (blue) and prediction of malignant area (red). The co-localization state was assigned to the spot when nine of the ten genes in the signature have expression values (UMI count > 0). The result of malignancy prediction is represented by the softmax value, with the likelihood of being malignant increasing as the value rises. Where 0.50 is the default softmax cutoff value. c, The proportions of the top ten representative cell types in spots

with a co-localization state across 5 slides. d, Gene expression of genes in the detected gene signature across cells. e, Survival analyses using the ssGESAs score of the genes of Hallmark EMT pathway. f, Survival analyses using the ssGESAs score of the gene signature from Cancer-Finder. g, Survival analyses using the ssGESAs score of the genes of ccRCC EMT program. Log-rank test was applied.

Line 382, Please delete (< 3000 words) from in front of the Methods.

Authors' response: We thank the reviewers for their careful reading. We have removed it.

Reviewer #5 (Remarks to the Author): Expert in single-cell RNA-seq and spatial transcriptomics methods and analysis, bioinformatics, and deep learning; co-reviewed with Reviewer #4

I thoroughly co-reviewed this manuscript with one of the reviewers who provided the listed reports as part of the Nature Communications initiative to facilitate training in peer review and appropriate recognition for co-reviewers.

Authors' response: We are grateful for the reviewer's positive comments on the novelty and significance of our study. In addition, we would like to thank the two reviewers for their insightful comments and assistance, which allowed us to strengthen this study. We have included additional rigorous cross-validation, detailed data preprocessing explanations, a comprehensive evaluation with multiple metrics, interpretability efforts, and sensitivity analysis in this version to address the issues. The detailed response is displayed above.

REVIEWER COMMENTS

Reviewer #1 (Remarks to the Author):

My comments have been adequately addressed.

Reviewer #2 (Remarks to the Author):

All my concerns have been addressed.

Reviewer #3 (Remarks to the Author):

I am satisfied with the changes made by the authors in regards to my review.

Reviewer #4 (Remarks to the Author):

Thank you for the submitted rebuttal letter. I would like to bring to your attention some aspects related to Figure 3 and the associated methodology that could benefit from further clarification.

1. External Validation in Figure 3: It is my understanding that Figure 3 is dedicated to the "external validation" of your model, utilizing datasets that were not included in the training set. However, the current presentation of this figure and its accompanying text raises certain questions regarding the methodology employed.

2. Repetition of Analysis: The manuscript indicates that the 10 datasets featured in Figure 3 were used for external validation. However, it's unclear why this analysis was repeated five times, as noted in Supplementary Tables 4-13. If the model has already been trained and is merely being evaluated on an external dataset, I would expect a single run to yield consistent results. This repetition suggests the

possibility of additional processes at play, such as fine-tuning on the test data, which should be explicitly addressed.

3. Clarification on Figures 1 and 2: While Figure 1 provides a useful overview and Figure 2 focuses on leave-cells-out cross-validation metrics, my primary interest lies in the methodology and results presented in Figure 3.

4. Interpretation of Multiple Training Runs: Upon reviewing the methods section, it appears that the model has been trained five times on the training data from TISCH, with each training run evaluated on the 10 external datasets. This approach, if confirmed, should be more clearly articulated in the figure caption. The current phrasing, "we ran the methods five times in parallel," lacks specificity and may lead to misinterpretation.

In light of these observations, I kindly request the following:

Confirmation and Clarification: Please confirm whether the model was indeed trained five times and evaluated separately on the external datasets. If this is the case, I recommend revising the figure caption to reflect this methodology more accurately and transparently. Please also apply this methodology to the method section of the manuscript.

Rationale for Multiple Runs: If multiple training and evaluation runs were conducted, a more detailed explanation of the rationale behind this approach would be valuable. Specifically, clarifying whether any form of fine-tuning or other modifications were applied to the model during these runs would be crucial for a complete understanding of your methodology.

I believe addressing these points will significantly enhance the clarity and robustness of your manuscript. I look forward to your response and any revisions you may find appropriate in light of these comments.

Reviewer #5 (Remarks to the Author):

I co-reviewed the revised manuscript together with Reviewer #4 and approve their opinion.

Response to Reviewers' Comments

Manuscript ID: NCOMMS-23-25528A

November 29, 2023

REVIEWER COMMENTS

Reviewer #1 (Remarks to the Author):

My comments have been adequately addressed.

Authors' response: We thank the reviewer for the positive comments.

Reviewer #2 (Remarks to the Author):

All my concerns have been addressed.

Authors' response: We thank the reviewer for the positive comments.

Reviewer #3 (Remarks to the Author):

I am satisfied with the changes made by the authors in regards to my review.

Authors' response: We thank the reviewer for the positive comments.

Reviewer #4 (Remarks to the Author):

Thank you for the submitted rebuttal letter. I would like to bring to your attention some aspects related to Figure 3 and the associated methodology that could benefit from further clarification.

Authors' response: Sincere thanks to the reviewers for the suggestions. We have considered the comments carefully and extended the description in Figure 3 accordingly. Please see our detailed response provided below.

1. External Validation in Figure 3: It is my understanding that Figure 3 is dedicated to the "external validation" of your model, utilizing datasets that were not included in the training set. However, the current presentation of this figure and its accompanying text raises certain questions regarding the methodology employed.

Authors' response: We appreciate the reviewers' comment. This understanding is absolutely correct, and we apologize for the lack of clarification.

Figure 3 illustrates the comparison between our model and other methods based on external validation. In our efforts to enhance the clarity of this section, we have made revisions to both the figure and its legend. The updated figure is presented below:

Revised Figure 3. Performance comparison with existing methods based on external validation datasets and application to a large database. a, Data structure of ten external validation datasets with varying cell counts and proportions of malignant cells. Among them, the head and neck dataset exclusively concentrated on immune cells, with malignant cells being experimentally excluded. b, Comparison of Cancer-Finder prediction accuracies to four other cell annotation algorithms on 10 external validation datasets. Cancer-Finder demonstrated significantly greater accuracies and similarities (to the labels from the original studies) than other tools across various cancers. Since most of these available algorithms exhibit some level of randomness in their results across runs, all tests were conducted in parallel five times. It is noteworthy that the pre-trained Cancer-Finder consistently yields uniform predictions on the external datasets. Recognizing that variations in the training process and data may introduce a degree of randomness, we conducted five training sessions for Cancer-Finder here, completing

the specified five repetitions (detailed in **Supplementary Note 2**). For ikarus (retrained), we employed the same strategy. The presence of an 'NA' denotes that the method returns an error and cannot be executed with these data. c, Comparison of Cancer-Finder's inference speed to the other four methods. 'NA' indicates that the method could not run correctly on the data. d, Comparison of Cancer-Finder's memory consumption to the other four methods. 'NA' indicates that the method could not run correctly on the data. This study utilized a maximum memory size of 512G Bytes. e, Up-regulated genes identified from predicted malignant cells. Full names of the cancers are detailed in **Supplementary Table 15**.

2. Repetition of Analysis: The manuscript indicates that the 10 datasets featured in Figure 3 were used for external validation. However, it's unclear why this analysis was repeated five times, as noted in Supplementary Tables 4-13. If the model has already been trained and is merely being evaluated on an external dataset, I would expect a single run to yield consistent results. This repetition suggests the possibility of additional processes at play, such as fine-tuning on the test data, which should be explicitly addressed.

Authors' response: We appreciate the reviewers' valuable comments and apologize for any confusion in our previous presentation.

We trained Cancer-Finder 5 times to complete the 5 repetitions here and did not make any fine-tuning to the test data. Specifically, we divided the training dataset from TISCH (328,230 cells) into 5 folds, following a similar approach to the internal dataset validation depicted in Figure 2. In each time, we trained the model using 4 out of the 5 folds and evaluated its performance on the external validation sets.

The decision to conduct multiple training is driven by two primary reasons. Firstly, since most of the other four algorithms exhibit some level of randomness in their results across runs, we ran them 5 times, to better reflect the randomness and accuracies of these methods. Secondly, although the predictions of a pre-trained Cancer-Finder on the external datasets remain consistent, its training process and training data introduce a degree of randomness that may result in fluctuations. Therefore, to comprehensively showcase the accuracies of Cancer-Finder, we performed multiple training to capture the potential range of its performance.

These details have been updated in **Supplementary Note 2** and **Figure 3** in this version.

3. Clarification on Figures 1 and 2: While Figure 1 provides a useful overview and Figure 2 focuses on leave-cells-out cross-validation metrics, my primary interest lies in the methodology and results presented in Figure 3.

Authors' response: We appreciate the reviewers' comment. As described in the previous comment, we conducted five training sessions for Cancer-Finder, utilizing 80% of the TISCH training data each time, and assessed its performance on the external validation sets. This approach aimed to account for potential randomness and showcase the comprehensive accuracies of Cancer-Finder.

4. Interpretation of Multiple Training Runs: Upon reviewing the methods section, it appears that the model has been trained five times on the training data from TISCH, with each training run evaluated on the 10 external datasets. This approach, if confirmed, should be more clearly articulated in the figure caption. The current phrasing, "we ran the methods five times in parallel," lacks specificity and may lead to misinterpretation. In light of these observations, I kindly request the following:

- **Confirmation and Clarification:** Please confirm whether the model was indeed trained five times and evaluated separately on the external datasets. If this is the case, I recommend revising the figure caption to reflect this methodology more accurately and transparently. Please also apply this methodology to the method section of the manuscript.

- **Rationale for Multiple Runs:** If multiple training and evaluation runs were conducted, a more detailed explanation of the rationale behind this approach would be valuable. Specifically, clarifying whether any form of fine-tuning or other modifications were applied to the model during these runs would be crucial for a complete understanding of your methodology.

Authors' response: We thank the reviewers' valuable advice. We trained Cancer-Finder 5 times to complete the 5 repetitions here and did not make any fine-tuning to the test data. As mentioned in response to comment #1 and 2, We have made revisions to Figure 3 and its legend, and additionally provided supplementary methods with explanations in the supplementary information section (**Supplementary Note 2: Model training for external validation**).

I believe addressing these points will significantly enhance the clarity and robustness of your manuscript. I look forward to your response and any revisions you may find appropriate in light of these comments.

Authors' response: We sincerely thank the reviewers' valuable feedback and suggestions, and we have explained and modified these points more clearly in this version.

Reviewer #5 (Remarks to the Author):

I co-reviewed the revised manuscript together with Reviewer #4 and approve their opinion.

Authors' response: Sincere thanks to the reviewer for the suggestions. We have considered the comments carefully and extended the description in Figure 3 and Supplementary **Note 2** accordingly.

REVIEWERS' COMMENTS

Reviewer #4 (Remarks to the Author):

I jointly reviewed the revised manuscript with Reviewer #5. The authors have addressed the issues we pointed out.

Reviewer #5 (Remarks to the Author):

REVIEWERS' COMMENTS

Reviewer #4 (Remarks to the Author):

I jointly reviewed the revised manuscript with Reviewer #5. The authors have addressed the issues we pointed out.

Authors' response: We thank the reviewer for the positive comments.

Reviewer #5 (Remarks to the Author):

Authors' response: We thank the reviewer for the positive comments.